# Ocular tropism of SARS-CoV-2 in animal models with retinal inflammation via neuronal invasion following intranasal inoculation

Gi Uk Jeong[1,13], Hyung-Jun Kwon[2,3,13], Wern Hann Ng[4,5,6], Xiang Liu[4,5,6], Hyun Woo Moon [1], Gun Young Yoon [1], Hye Jin Shin[1,12], In-Chul Lee[2,3], Zheng Lung Ling[7,8], Alanna G. Spiteri[7,8], Nicholas J. C. King [4,7,8,9,10], Adam Taylor[4,5,6], Ji Soo Chae[11], Chonsaeng Kim[1], Dae-Gyun Ahn [1], Kyun-Do Kim[1], Young Bae Ryu[2], Seong-Jun Kim [1], Suresh Mahalingam [4,5,6,14] & Young-Chan Kwon [1,14] ✉

Although ocular manifestations are reported in patients with COVID-19, consensus on ocular tropism of SARS-CoV-2 is lacking. Here, we infect K18-hACE2 transgenic mice with SARS-CoV-2 using various routes. We observe ocular manifestation and retinal inflammation with production of pro-inflammatory cytokines in the eyes of intranasally (IN)-infected mice. Intratracheal (IT) infection results in dissemination of the virus from the lungs to the brain and eyes via trigeminal and optic nerves. Ocular and neuronal invasions are confirmed using intracerebral (IC) infection. Notably, the eye-dropped (ED) virus does not cause lung infection and becomes undetectable with time. Ocular and neurotropic distribution of the virus in vivo is evident in fluorescence imaging with an infectious clone of SARS-CoV-2-mCherry. The ocular tropic and neuroinvasive characteristics of SARS-CoV-2 are confirmed in wild-type Syrian hamsters. Our data can improve the understanding regarding viral transmission and clinical characteristics of SARS-CoV-2 and help in improving COVID-19 control procedures.

Several respiratory viruses, such as species D adenoviruses and subtype H7 influenza viruses, show ocular tropism and cause ocular diseases in humans[1]. While ocular manifestations and abnormalities have commonly been reported in patients with COVID-19[2,3], the ocular

tropism and pathologies of SARS-CoV-2 remain unclear. Viral RNA was not detected using reverse transcription-quantitative PCR (RT-qPCR) analysis of the conjunctival swabs of three patients with COVID-19 and bilateral conjunctivitis[4] and in 16 aqueous humour and vitreous

[1]Department of Convergent Research of Emerging Virus Infection, Korea Research Institute of Chemical Technology, Daejeon 34114, Republic of Korea. [2]Department of Functional Biomaterial Research Center, Korea Research Institute of Bioscience and Biotechnology, Jeongeup 56212, Republic of Korea. [3]Center for Companion Animal New Drug Development, Jeonbuk Branch, Korea Institute of Toxicology, Jeongeup 53212, Republic of Korea. [4]Emerging Viruses, Inflammation and Therapeutics Group, Menzies Health Institute Queensland, Griffith University, Gold Coast, QLD, Australia. [5]Global Virus Network (GVN) Centre of Excellence in Arboviruses, Griffith University, Gold Coast, QLD, Australia. [6]School of Pharmacy and Medical Sciences, Griffith University, Gold Coast, QLD, Australia. [7]Viral Immunopathology Laboratory, The Charles Perkins Centre, The University of Sydney, Sydney, NSW 2006, Australia. [8]School of Medical Sciences, Faculty of Medicine and Health, The University of Sydney, Sydney, NSW 2006, Australia. [9]Sydney Institute for Infectious Diseases, The University of Sydney, Sydney, NSW 2006, Australia. [10]Sydney Nano, The University of Sydney, Sydney, NSW 2006, Australia. [11]Department of Life Sciences, PerkinElmer, Seoul 08380, Republic of Korea. [12]Present address: Department of Microbiology, Chungnam National University School of Medicine, Daejeon 35015, Republic of Korea. [13]These authors contributed equally: Gi Uk Jeong, Hyung-Jun Kwon. [14]These authors jointly supervised this work: Suresh Mahalingam, Young-Chan Kwon. ✉e-mail: yckwon@krict.re.kr

samples from post-mortem cases[5]. In contrast, a cross-sectional study in Italy reported the presence of viral RNA in the conjunctival swabs of 52 of 91 patients with COVID-19, with a high rate of detection in patients with severe disease[6]. Other cross-sectional studies in China[7] and Brazil[8], and post-mortem examinations[9,10] also reported viral RNA detection in conjunctival swabs or in the aqueous humour of patients with or without ocular manifestations.

As the ocular surface represents an additional mucosal surface exposed to infectious aerosols, the eyes are considered a potential transmission route of SARS-CoV-2. Indeed, the SARS-CoV-2 receptor is expressed on the human ocular surface[11,12]. Deng et al. showed that SARS-CoV-2 potentially entered via the conjunctiva in rhesus macaques after conjunctival inoculation[13]. In addition, SARS-CoV-2 can infect human embryonic stem cell-derived ocular epithelial cells[14] and human conjunctival epithelial cells[15]. Thus, whether the eyes can act as primary or secondary virus entry sites needs to be investigated to guide preventative or therapeutic strategies against SARS-CoV-2 transmission.

In this work, we evaluate the ocular tropism and the possible ocular transmission of SARS-CoV-2 using K18-hACE2 transgenic mice and wild-type Syrian hamsters. Using various inoculation routes, we demonstrate the ocular tropism of SARS-CoV-2 via neuronal invasion of trigeminal (TN) and optic nerves (ON) in the mouse model. Using infectious SARS-CoV-2-mCherry clones and a fluorescence imaging system, we examine the ocular and neurotropic distribution of the virus in vivo. In addition, we provide evidence for the ocular tropic and neuroinvasive characteristics of SARS-CoV-2 in wild-type hamsters.

## Results

### Detection of SARS-CoV-2 and ocular manifestations in the eyes of infected mice

A previous study has reported that Zika virus (ZIKV) infection of the eyes led to the development of conjunctivitis and panuveitis, and caused inflammation in the uveal tissues of a mouse model[16]. To investigate whether the pathogenesis of the ocular disease is caused in response to the SARS-CoV-2 infection in K18-hACE2 mice, we intranasally (IN) infected mice with $10^4$ plaque-forming units (PFU) of the virus or the same volume of phosphate-buffered saline (PBS) (mock group). Consistent with previous findings[17], we observed 20% weight loss at 7 days post-infection (dpi; Fig. 1a) and mortality at 8 dpi (Fig. 1b). Notably, at 6 dpi, tearing and eye discharge occurred in 25% of infected mice (Fig. 1c). We then evaluated the presence of SARS-CoV-2 in the eyes of mice after IN infection. At 6 dpi, the viral load in the lungs and eyes, including appendages, was assessed by plaque assay. The infectious viral titre obtained from the eyes was as high as that from the lung (~10^6 PFU/g; Fig. 1d). Considering the detection of viral RNAs in tears from patients with SARS-CoV[18] and SARS-CoV-2[19], we assessed the viral RNA copies in the lacrimal gland at 6 dpi by RT-qPCR targeting the viral nucleocapsid gene (Fig. 1e). Interestingly, the viral load of the lacrimal gland was significantly lower than that of the eyes and was similar to that in the spleen (~10^4 viral RNA copies per microgram of total RNA), one of the tissues that is negligibly susceptible to the infection[20]. These findings were also confirmed in female K18-hACE2 mice, indicating that the virus lacked sex- or gender-specificity (Supplementary Fig. 1). We next examined the retinal sections of IN-infected mice at 6 dpi for detecting viral spike (S) protein using immunofluorescence staining. The ACE2 protein was observed in the retina (Supplementary Fig. 2). The S protein was mostly observed in the ganglion cell layer of the retina that consists of retinal ganglion cells, the axons of which form the optic nerve, chiasma, and tract (Fig. 1f). We also observed the co-localisation of the S protein and γ-synuclein, a marker of retinal ganglion cells[21], in the ganglion cell layer, indicating that the infected cells were possibly retinal ganglion cells (Supplementary Fig. 3). These results demonstrated the presence of infectious SARS-CoV-2 in the eyes, suggesting that the eyes were the target of SARS-CoV-2 infection possibly via neuronal invasion.

### Retinal inflammation and cytokine production in the eyes of SARS-CoV-2-infected mice

We further examined the histopathological and inflammatory changes in response to the infection using haematoxylin-eosin (H&E)-stained eye sections from mock- or SARS-CoV-2 IN-infected mice. As retinal thickness is considered a potential inflammatory signature[22], we measured retinal thickness following infection to determine retinal inflammation. Compared to that observed in the mock group, lesions in retinal tissues were observed in the SARS-CoV-2-infected mice (Fig. 2a and Supplementary Fig. 4). The mean retinal thickness (from the ganglion cell layer to the outer-nuclear layer) was 46.27 μm in the mock group and 75.14 μm in the infected group (Fig. 2b). Viral infection significantly increased the retinal thickness by 1.62-fold, inducing substantial accumulation of infiltrating inflammatory cells in the ganglion cell, inner-, and outer-nuclear layers. To identify the infiltrating immune cells, we stained the eyes for $CD3^+$ T, $CD4^+$ T and $CD8^+$ T cells, macrophages/inflammatory monocytes, and $GR1^+$ neutrophils using immunofluorescence staining (Fig. 2c and Supplementary Fig. 5). Higher T cell and neutrophil levels were observed in the eyes of SARS-CoV-2-infected mice at 6 dpi than that in mock-infected mice. Furthermore, at 6 dpi, the multiplex immuno-analysis of the eyes, including appendages, showed elevation in the levels of pro-inflammatory cytokines and chemokines, including granulocyte colony-stimulating factor (G-CSF), interferon gamma-inducible protein-10 (IP-10), MKC, monocyte chemoattractant protein-1 (MCP-1), macrophage-inflammatory protein-2 (MIP-2), interleukin (IL)−6 and IL-12, in response to the infection (Fig. 3a). The levels of pro-inflammatory cytokines and chemokines were significantly augmented in the brain compared to the lung, where the viral load was comparable to that in the eyes, including appendages (Fig. 1e and Supplementary Fig. 6). These findings indicated that IN-administered SARS-CoV-2 promoted retinal inflammation and production of pro-inflammatory cytokines and chemokines in the eyes of K18-hACE2 mice.

Next, we evaluated the functional consequences of retinal inflammation on visual behaviour. The innate aversion to depth has been exploited in the visual cliff test designed to study depth perception in mice using a visual cliff apparatus shown in Fig. 3b[23]. The test order was randomised and each mouse was tested only once in a lifetime to avoid a memory effect. At 5 dpi, the SARS-CoV-2-infected mice were divided into two groups based on the presence or absence of ocular symptoms. On the same day, the mice of mock and SARS-CoV-2-infected groups dismounted the platform within 4.28 and 3.56 s (mean), respectively (Fig. 3c). The SARS-CoV-2-infected mice with ocular symptoms presented prolonged latency for the dismounting (42.92 s). However, the number of mice with first foot on the cliff side did not differ between mock and SARS-CoV-2-infected groups regardless of ocular symptoms (Fig. 3d), indicating that the retinal inflammation induced by viral infection did not exacerbate retinal degeneration or visual loss.

### Ocular tropism of SARS-CoV-2 via neuronal invasion in mice

Similar to other neurotropic viruses, such as ZIKV and West Nile viruses, SARS-CoV and SARS-CoV-2 have been reported to cause symptoms of neurological disorders in patients[24,25]. Recently, evidence of rapid SARS-CoV-2 spread to the olfactory bulb via olfactory nerves was detected in patients with COVID-19[26,27]. As both the olfactory nerve and TN provide anatomical connections between the brain and nasal passages[28], we hypothesised that IN-administered SARS-CoV-2 spreads to the brain and eyes via the TN and ON (Fig. 4a). To validate this hypothesis, we infected mice with SARS-CoV-2 using various injection routes (Fig. 4b; IT: intratracheal, IC: intracerebral, ED: eye-drop, IV: intravenous). A dose of $10^4$ PFU was injected into the mice and viral loads were assessed in the lungs, brain, eye globes, TN and ON at 3 and 6 dpi. Overall weight loss in mice injected via the IN, IT, and IC routes was observed. Mortality was observed only in mice injected via the IC

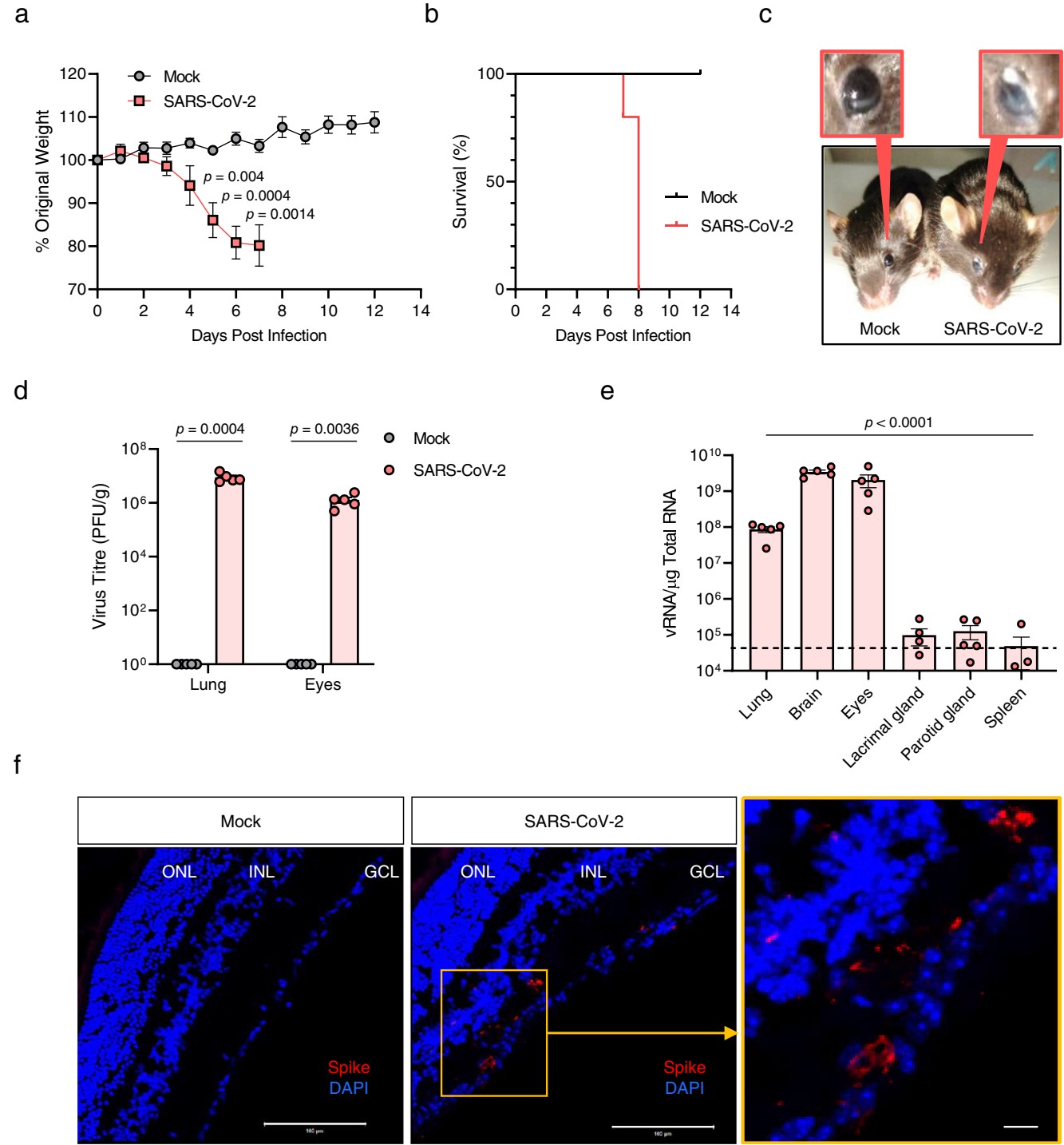

**Fig. 1 | Clinical features and virus titres in the eyes of SARS-CoV-2-infected mice.** Eight- to nine-week-old male K18-hACE2 mice were intranasally mock-infected or infected with $10^4$ PFU of SARS-CoV-2 (n = 5 for mock- and SARS-CoV-2-infected mice, respectively; Mock, Grey; SARS-CoV-2, Red). **a** Body weight changes shown as percentage of starting weight. **b** Survival was evaluated at the indicated dpi. **c** Representative image of tearing and eye discharge in SARS-CoV-2-infected mice at 6 dpi (right) compared to those in mock-infected mice (left). **d** Viral load in the lungs and eyes, including appendages, was analysed using a plaque assay at 6 dpi. **e** Viral RNA levels in the lungs, brain, eyes, including appendages, lacrimal gland, parotid gland and spleen were assessed using RT-qPCR at 6 dpi. Viral RNA copies

were cut-off ($10^4$ copies/µg). A dashed line indicates the viral RNA levels of the spleen as a limit of detection. **f** Representative confocal images of immuno-fluorescence stained retinal sections of IN-infected mice (n = 3 per group) for viral spike (S) protein (red) at 6 dpi. DAPI staining (blue) was used to visualise nuclei of the ganglion cell layer (GCL), inner nuclear layer (INL), and outer-nuclear layer (ONL) in the retinal cross-section. Scale bar = 100 µm. Symbols represent means ± SEM. Statistically significant differences between the groups were determined using multiple two-tailed t-tests (**a**), unpaired two-tailed t-test (**d**) or one-way ANOVA (**e**). SARS-CoV-2: severe acute respiratory syndrome coronavirus 2; PFU plaque-forming unit, vRNA viral RNA, dpi days post-infection.

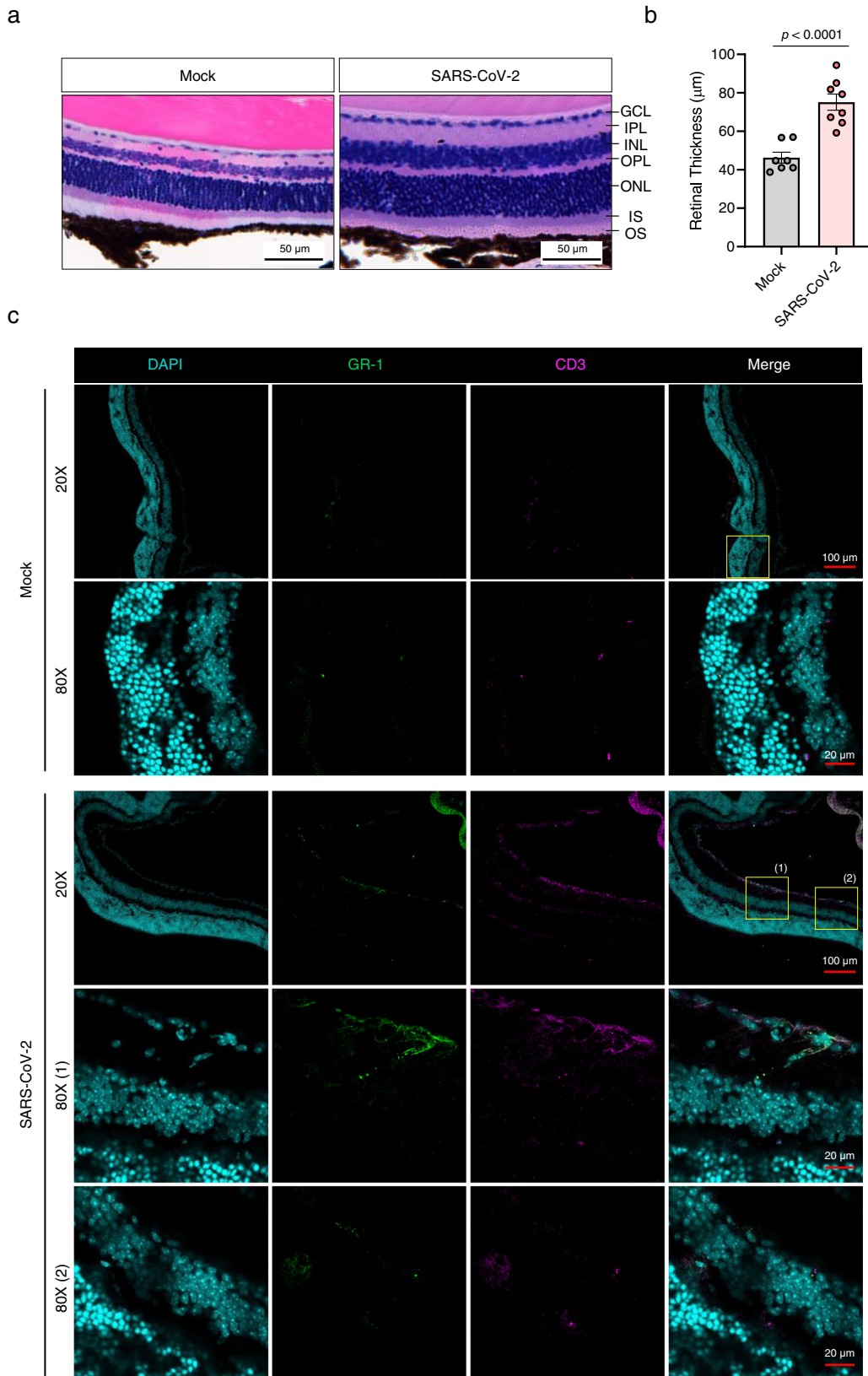

route from 2 dpi (Supplementary Fig. 7). This observation is consistent with that of a previous study that demonstrated that brain infection of SARS-CoV leads to the death of K18-hACE2 mice[29]. The IN injection resulted in ocular tropism with increased viral RNA levels in the eye globes between 3 and 6 dpi (Fig. 4c). In general, the viral RNA titres detected in the TN and ON were comparable to those in the brain and eye globes, indicating viral transmission to the eyes and brain via neuronal invasion of these nerves. We also detected infectious viral particles in the TN and ON, the pattern of which was similar to that of the viral RNA levels (Supplementary Fig. 8). To further investigate this dissemination from the lung to the eyes and brain, we directly injected the virus into the lungs using the IT route. In a similar manner to IN

**Fig. 2 | Histopathological and immunofluorescence analyses of the eyes of SARS-Cov-2-infected mice. a** H&E staining of the eye sections from K18-hACE2 mice six days post mock infection or SARS-CoV-2 infection ($10^4$ PFU). Representative histological images ($n = 4$ per group) show changes in the retinal thickness. Scale bar = 50 μm. **b** The retinal thickness ($n = 4$ per group, total eight eyes) was measured and shown in a bar graph (Mock, Grey; SARS-CoV-2, Red). Symbols represent means ± SEM. Statistically significant differences between the groups were determined using an unpaired two-tailed *t*-test. **c** Immunofluorescence analysis of eye tissues from SARS-CoV-2-infected mice (mock, $n = 8$; SARS-CoV-2,

$n = 10$) at 6 dpi. Cryosections were labelled for Gr-1 (neutrophils), CD3 (T cells), and DAPI (nucleus). The differences in the infiltration of Gr-1 neutrophils and CD3+ T cells between mock- and SARS-CoV-2-infected mice are shown in the images. The confocal microscopy images were acquired using a 20× and 40× (with 2× zoom) objective. Scale bars in panel = 100 μm for 20× objective; 20 μm for 40× (with 2× zoom) objective. Data were representative of two independent experiments. SARS-CoV-2 severe acute respiratory syndrome coronavirus 2, GCL ganglion cell layer, IPL inner plexiform layer, INL inner nuclear layer, OPL outer plexiform layer, ONL outer-nuclear layer, IS inner segments, OS outer segments, dpi days post-infection.

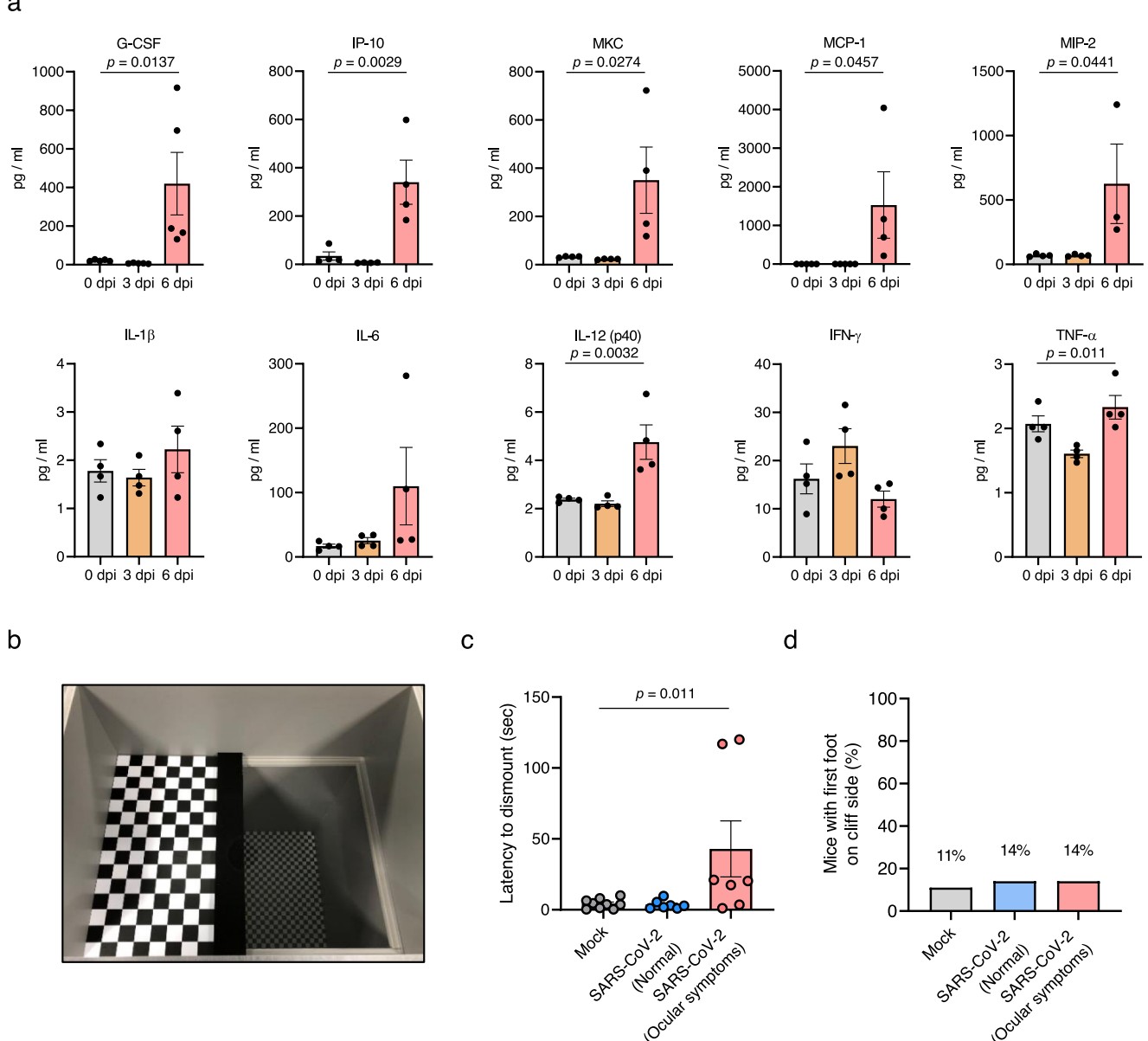

**Fig. 3 | Multiplex cytokine analysis of the eyes and the visual cliff test for SARS-Cov-2-infected mice. a** The chemokine and cytokine levels in the eyes, including appendages, in SARS-CoV-2-infected mice, were measured using multiplex immuno-analysis ($n = 4$ per indicated dpi; 0 dpi, grey; 3 dpi, yellow; 6 dpi, red). G-CSF granulocyte-macrophage colony-stimulating factor, IP-10 C-X-C motif chemokine 10 (CXCL10), MKC mouse keratinocyte-derived chemokine, MCP-1 monocyte chemoattractant protein-1 (CCL2), MIP-2 macrophage-inflammatory protein-2

(CXCL2). **b** Photo of the visual cliff apparatus used in this study. **c, d** Time for latency to dismount platform (**c**) and the percentage of mice that took the first footstep over the cliff side (**d**) were measured (Mock, $n = 9$; SARS-CoV-2 (Normal), $n = 7$; SARS-CoV-2 (Ocular symptoms), $n = 7$). Symbols represent means ± SEM. Statistically significant differences between the groups were determined using one-way ANOVA (**a, c**). SARS-CoV-2 severe acute respiratory syndrome coronavirus 2, dpi days post-infection.

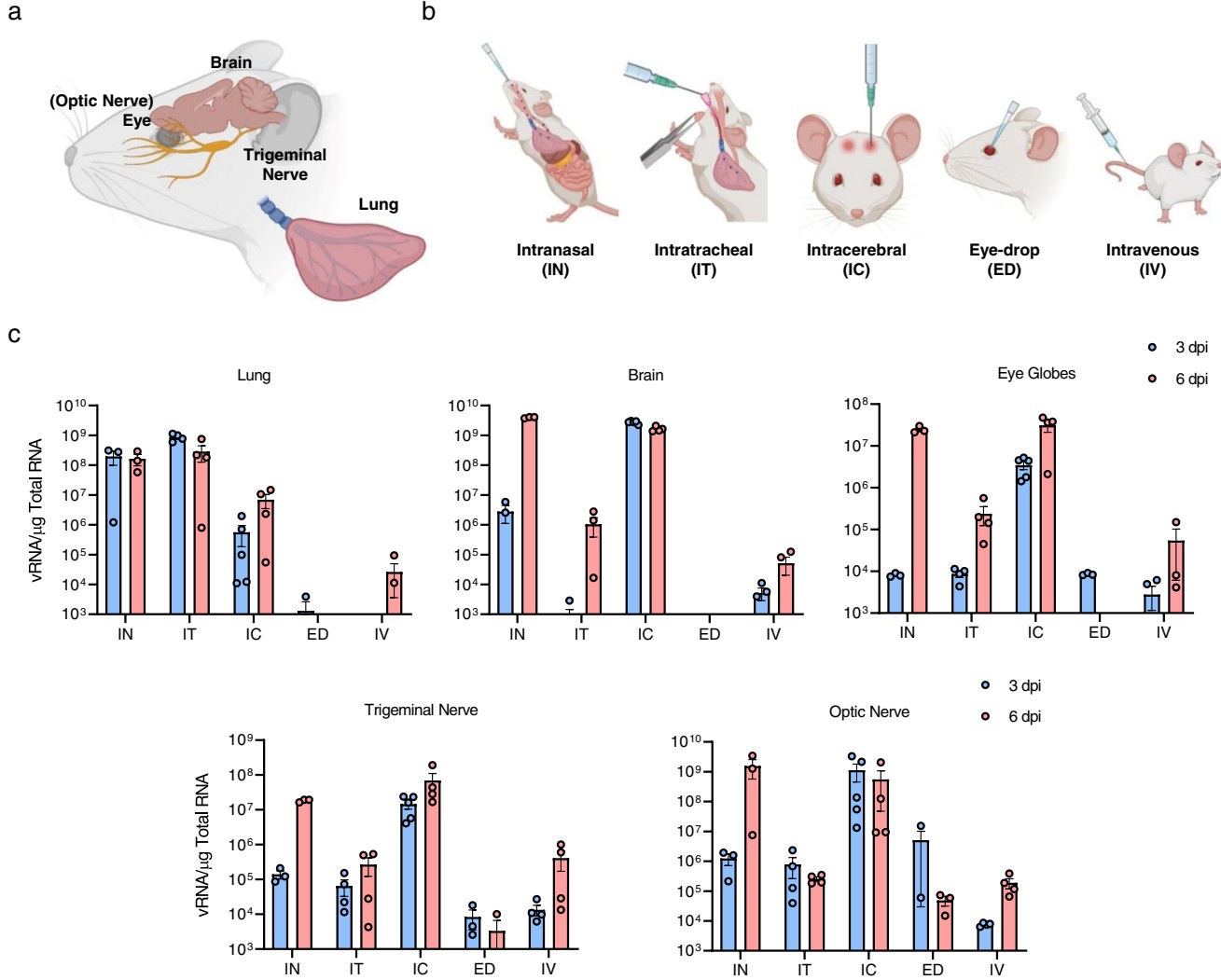

**Fig. 4 | Virus titres of SARS-CoV-2-infected mice via diverse injection routes.** **a** An anatomical graphic depicting the murine brain, eye (optic nerve), trigeminal nerve and lung. Created with BioRender.com. **b** Illustration showing the diverse injection routes: intranasal (IN), intratracheal (IT), intracerebral (IC), eye-drop (ED) and intravenous (IV) routes. Created with BioRender.com. **c** K18-hACE2 mice were inoculated with $10^4$ PFU SARS-CoV-2 via five different injection routes (n = 8 per injection route; n = 4 for 3 dpi and 6 dpi, respectively). Viral RNA levels in the lungs, brain, eye globes, trigeminal nerve and optic nerve were analysed at 3 and 6 dpi using RT-qPCR (3 dpi, blue; 6 dpi, red). Viral RNA copies were subjected to cut-off at $10^3$ copies/μg. Symbols represent means ± SEM. SARS-CoV-2 severe acute respiratory syndrome coronavirus 2, PFU plaque-forming unit; dpi: days post-infection.

infection, increased viral RNA titres were also detected in the eye globes, brain, TN, and ON at 6 dpi, indicating viral transmission from the lung to the eyes and brain via TN and ON. To confirm the infection route between the eyes and brain via TN and ON, we directly injected the virus into the brain via the IC route. Viral RNA copies were found in the eye globes, brain, TN and ON; however, the copies of the viral RNA in the lung were relatively lower than those in other tissues. IC infection evidently showed that the virus can migrate to the eyes through the TN and ON from the brain. In addition, as both IN and IC infections showed the highest viral titre in the eyes at 6 dpi, tearing and eye discharge were found only in mice infected via the IN and IC routes. These results revealed that viral transmission occurs between the brain and eyes via the TN and ON, with a network reaching the lung.

The ocular surface of human[12,14] and murine models[30], which expresses ACE2 and TMPRSS2, is considered an additional mucosal surface possibly exposed to virus-containing aerosols. To investigate whether the virus can enter the eyes and migrate to the respiratory tract, we measured the viral load following the ED infection route (dose of $10^4$ PFU). The viral RNA load was detected in the eye globes at 3 dpi, but not at 6 dpi, and low viral load was detected only in the TN and ON.

This demonstrated the faint possibility of ocular susceptibility to SARS-CoV-2. A previous study has suggested that SARS-CoV-2 infection via the IV route was not required for lung infection[31]. Indeed, the IV infection resulted in relatively low viral copies in every tissue studied, which included the lungs, brain, eye globes, TN and ON. Overall, SARS-CoV-2 can migrate from the respiratory tract to the eyes through the brain via TN and ON, which cannot be reversed.

## Ocular- and neuro-tropism of infectious SARS-CoV-2-mCherry in mice

The incorporation of a fluorescent reporter gene into a replication-competent virus has advanced our ability to trace viral infection and tropism in vivo. To confirm the viral spread from the respiratory tract to eyes through the brain, we used an infectious clone, SARS-CoV-2-mCherry, generated using the reverse genetics system[32]. A sequence encoding the mCherry fluorescent marker was inserted in-frame after the C-terminus of the ORF7a protein to avoid the deletion of viral sequences. This virus can replicate and recapitulate features of severe COVID-19 infections associated with the mouse model[33]. In mice infected via the IN route with $10^4$ PFU of SARS-CoV-2-mCherry,

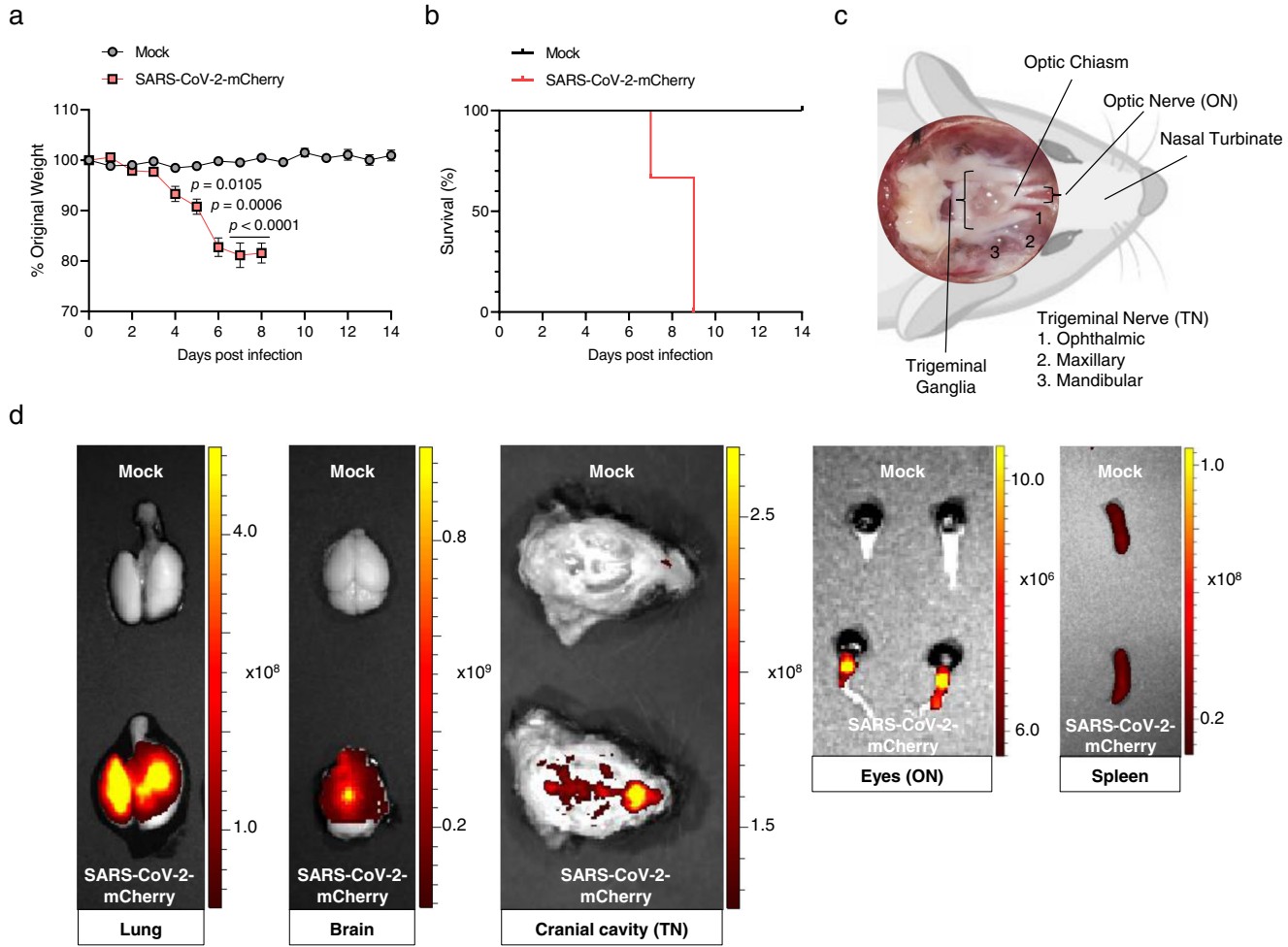

**Fig. 5 | In vivo viral distributions of infectious SARS-CoV-2-mCherry in IN-infected mice.** K18-hACE2 mice were IN-inoculated with $10^4$ PFU of SARS-CoV-2-mCherry ($n = 6$ for the mock-infected and infected mice, respectively). **a**, **b** Body weight (**a**) and survival (**b**) were monitored at the indicated dpi (Mock, grey; SARS-CoV-2-mCherry, red). Symbols represent means ± SEM. Statistically significant differences between the groups were determined using multiple two-tailed *t*-tests. **c** A brain-dissected dorsal view of the murine cranial cavity showing optic nerves, optic chiasma, trigeminal ganglia and trigeminal branches (1, ophthalmic; 2, maxillary; 3, mandibular). Created with BioRender.com. **d** Representative in vivo fluorescence images of organs, including the lungs, brain, eyes and spleen in mock- or SARS-CoV-2-mCherry-infected mice at 6 dpi ($n = 5$ for the mock-infected and infected mice, respectively; upper, mock; lower, SARS-CoV-2-mCherry) were acquired using sequential imaging. Colour bars indicate radiant efficiency [p/sec/cm$^2$/sr]/[μW/cm$^2$]. SARS-CoV-2 severe acute respiratory syndrome coronavirus 2, PFU plaque-forming unit; dpi: days post-infection.

significant morbidity was observed from 5 dpi (Fig. 5a). Mortality was observed from 7 dpi to 9 dpi (Fig. 5b). The remaining mice met humane endpoint criteria at 9 dpi. The viral distribution in tissues, including the lungs, brain, eye globes, spleen, TN, and ON was analysed at 6 dpi using an in vivo imaging system to detect the fluorescence of SARS-CoV-2-mCherry. For detecting the fluorescence in the TN and ON of euthanized mice, we cut the parietal bone along the left and right sides and the sagittal suture, followed by the removal of the brain gently to expose the base of the cranial cavity (Fig. 5c). Robust fluorescence signals were detected in every tissue studied (ranged from $10^6$ to $10^9$ radiant efficiency [p/sec/cm$^2$/sr]/[μW/cm$^2$]), which were clearly distinguishable from that of the mock group, except for that in the spleen, the negative control (Fig. 5d). The viral distribution to the brain and eyes validated the neuronal invasion of the virus to the TN and ON that can be used for SARS-CoV-2 transmission.

## Ocular tropism of SARS-CoV-2 through the brain via the TN and ON in wild-type Syrian hamsters

As the expression and distribution of hACE2 in K18-hACE2 mice are under the control of the cytokeratin 18 promoter[34], these animals are not naturally sensitive to SARS-CoV-2 infection. Investigation of the

ocular tropism in wild-type Syrian hamsters, the endogenous ACE2 of which can interact with SARS-CoV-2 to make it naturally permissive to viral infection[35–38], needs to be extrapolated to humans. SARS-CoV-2 IN-inoculation (dose of $10^4$ PFU) of Syrian hamsters resulted in weight loss (~10%) at 6 dpi, although most animals returned to their original weight by 14 dpi (Fig. 6a). During the course of the experiments, no mortality was observed in either mock or infection groups. To determine the ocular tropism of SARS-CoV-2 in a hamster model, we infected the virus via IN and ED routes. Compared to that observed with IN infection, no significant weight loss was observed for ED-infected (dose of $10^4$ PFU) hamsters (Fig. 6b). The quantity of infectious SARS-CoV-2 in the lungs along with the eye globes at 6 days after IN or ED infection was assessed by plaque assay. Infectious virus in the eye globes was detected in the IN-infected hamsters, confirming the ocular tropism of SARS-CoV-2 in wild-type animals (Fig. 6c, d). Furthermore, analysis of virus titre using RT-qPCR showed the presence of the virus in other tissues of IN-infected hamsters, including the brain, ON, and TN at titres higher than that in the spleen, which is negligibly susceptible to SARS-CoV-2 infection (Fig. 6e). However, marginally higher viral RNA levels were detected only in the eye globes and ON of ED-infected hamsters than that in the spleen (Fig. 6f). Consequently,

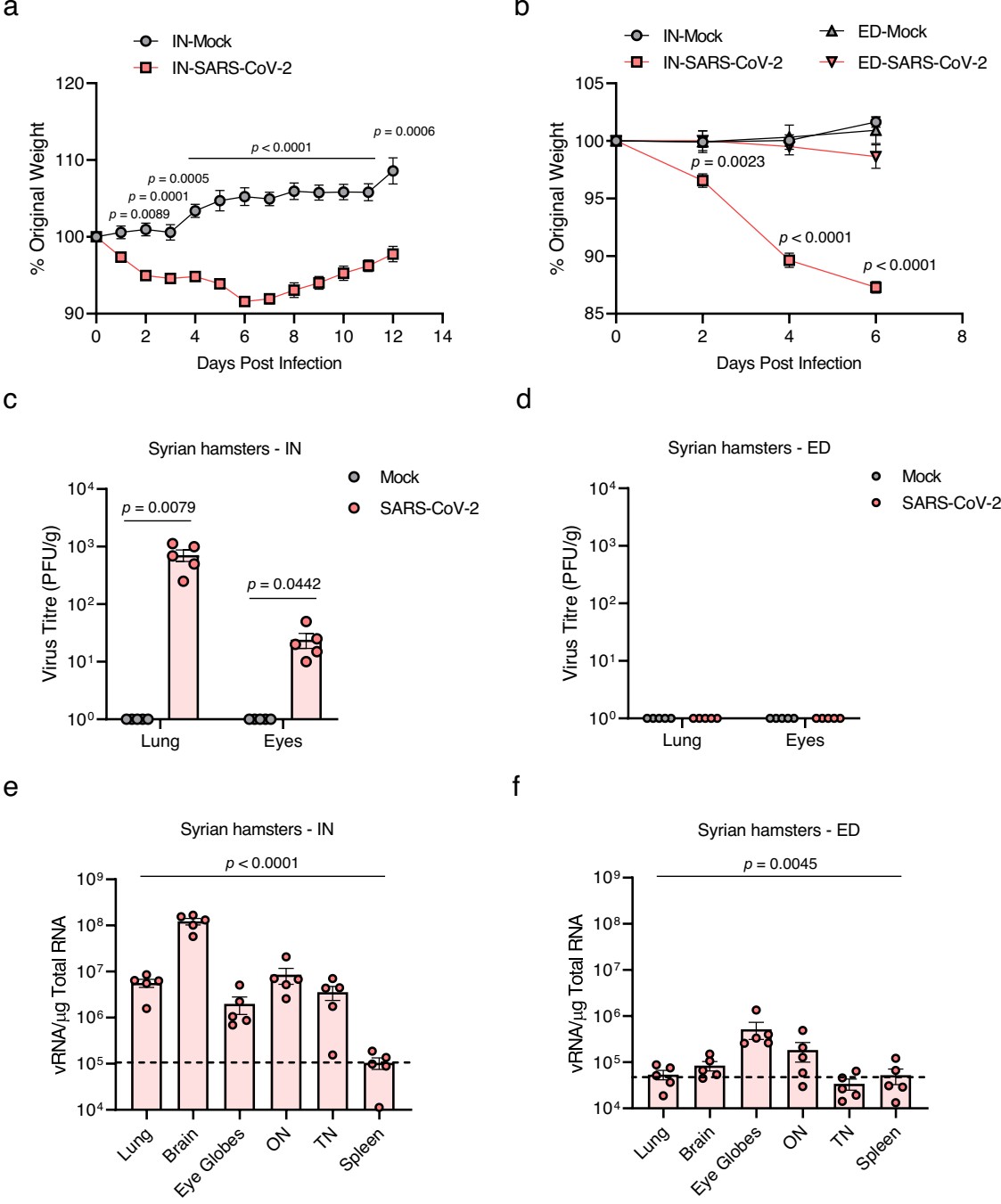

**Fig. 6 | Clinical features and viral titres of SARS-CoV-2 in intranasally (IN) or eye-drop (ED)-infected wild-type Syrian hamsters.** Eleven-week-old female Golden Syrian hamsters were IN-mock-infected or infected with $10^4$ PFU of SARS-CoV-2 ($n = 6$ for mock-infected and $n = 5$ for infected mice; mock, grey; SARS-CoV-2, red). **a** Body weight changes are shown as the percentage of the starting weight at the indicated dpi after IN infection. **b** Syrian hamsters were IN- or ED-infected with $10^4$ PFU of SARS-CoV-2 (each $n = 5$ for IN-mock-infected, IN-infected, ED-mock-infected, and ED-infected). A graph showing the percent body weight change is shown. **c, d** Viral loads at 6 dpi in the lungs and eyes of IN-infected (**c**) and ED-infected (**d**) hamsters were measured using plaque assay. **e, f** Viral RNA levels at 6 dpi in the lungs, brain, eye globes, optic nerves (ONs), trigeminal nerves (TNs) and spleens of IN-infected (**e**) and ED-infected (**f**) animals were assessed using RT-qPCR. Viral RNA copies were subjected to a cut-off at $10^4$ copies/µg. The dashed line indicates the viral RNA levels of the spleen as a limit of detection. Symbols represent means ± SEMs. Statistically significant differences between the groups were determined using multiple two-tailed t-tests (**a, b**), unpaired two-tailed *t*-test (**c, d**), or one-way ANOVA (**e, f**). SARS-CoV-2 severe acute respiratory syndrome coronavirus 2, PFU plaque-forming unit, dpi days post-infection.

analysis of gross pathology revealed pathological lesions and pulmonary congestion only in IN-SARS-CoV-2-infected hamsters at 6 dpi (Fig. 7a). Histopathological changes such as thickening of alveolar septum and infiltration of inflammatory cells in the lungs were also observed only in IN-SARS-CoV-2-infected hamsters (Fig. 7B). These data are consistent with the aforementioned findings in K18-hACE2 mice, suggesting that SARS-CoV-2 can spread from the respiratory tract to the brain and eyes via the TN and ON, which cannot be reversed.

## Discussion

A previous study has highlighted the neuro-tropism of SARS-CoV-2 and suggested the need to identify the route of viral invasion into the brain[39]. Olfactory nerves form bundles that provide an anatomical

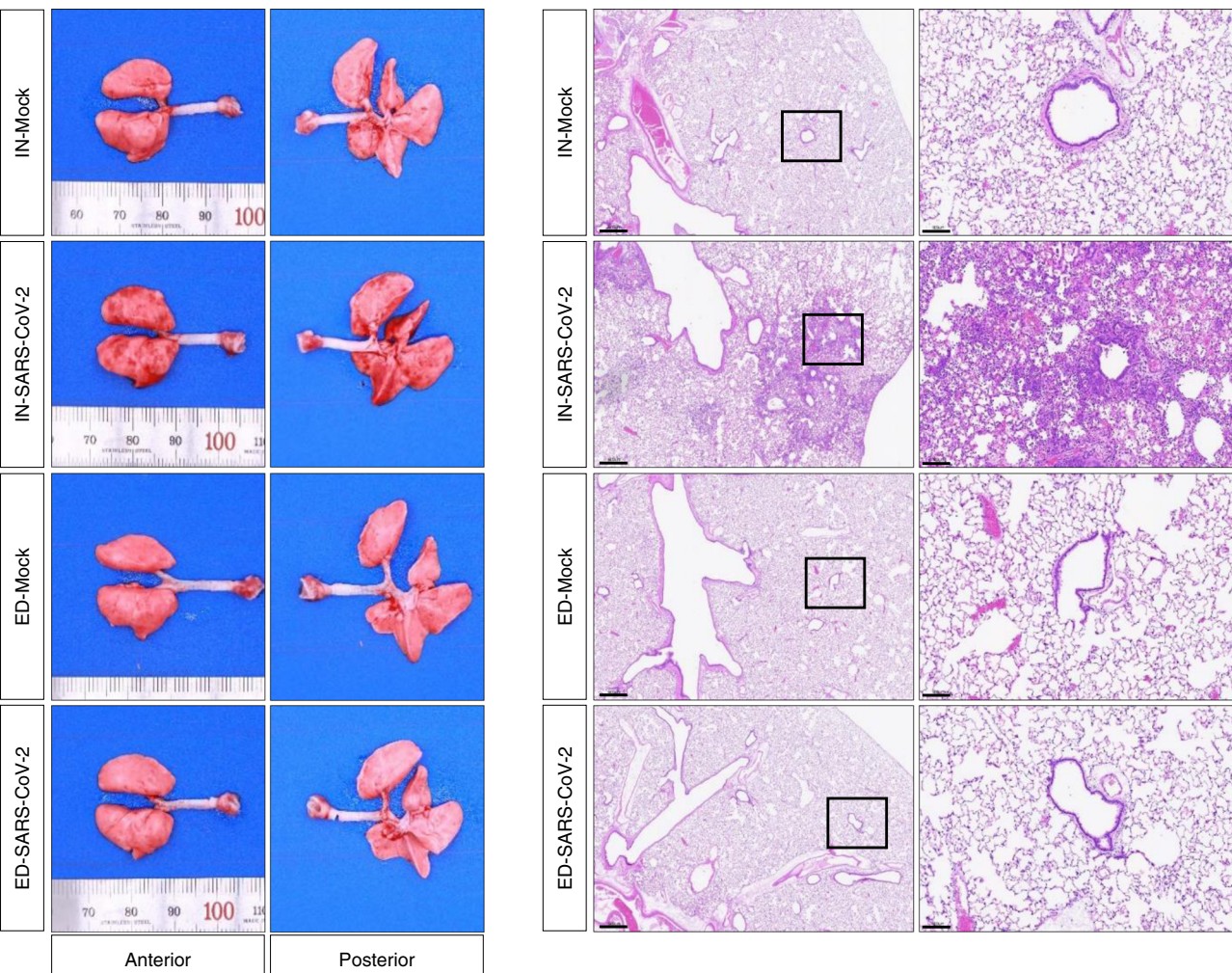

**Fig. 7 | Gross and pulmonary histopathology of intranasally (IN) or eye-drop (ED)-infected wild-type Syrian hamsters.** Syrian hamsters were IN- or ED-infected with $10^4$ PFU of SARS-CoV-2 (each $n = 5$ for IN-mock-infected, IN-infected, ED-mock-infected, and ED-infected) **a** Representative digital image of lungs harvested from mock-infected or infected hamsters at 6 dpi. **b** Representative H&E stained lung tissue slices of mock-infected or infected hamsters at 6 dpi used to identify histopathological changes. The left panels represent low-magnification images (scale bar = 500 μm). The right panels show high-magnification images of the regions designated by squares (scale bar = 100 μm).

connection between the brain and nasal passage through the foramina in the cribriform plate and synapse on the glomeruli in the olfactory bulb[40]. In addition, branches $V_1$ (ophthalmic) and $V_2$ (maxillary) of TN (Fig. 5c) innervate both the respiratory and olfactory regions of the nasal passage, connecting the brain[41,42]. Although SARS-CoV-2 infection and transmission via the olfactory nerves have been demonstrated by several research groups[26,27,43], those of TN remains to be elucidated. Based on data obtained from a patient with COVID-19 who suffered from trigeminal neuralgia, neuronal invasion via TN has been suggested in humans[44]. In this study, we demonstrated that TN can be infected by SARS-CoV-2 and used for transmitting the virus to the brain and eyes, along with the ON.

As the nasolacrimal duct provides an anatomical connection between the ocular surface and respiratory tract[45], the spreading of the virus to the eyes could have occurred via the nasolacrimal duct, and not the neurons. Although we used IT infection to prevent early transmission to the eyes via the nasolacrimal duct (Fig. 4), post-inoculation progeny viruses may move toward the eyes via the naso-lacrimal duct following viral replication in the lungs, which has not been investigated in this study. Further studies on viral replication in the upper respiratory tract under conditions that stop spreading to the eyes via the nasolacrimal duct are warranted.

A recent clinical study revealed that patients with COVID-19 showed lesions at the ganglion cell layer and inner plexiform layer in both eyes[46]. In addition, another clinical study reported that the patients presented flame-shaped haemorrhages along the retinal vascular arcades and peripheral retinal haemorrhages[47]. The breakdown of the blood-retinal barrier can cause immune cells infiltration and abnormal fluid accumulation within the retina following an increase in retinal thickness[48]. In this study, we have observed immune cells infiltration in the inner nuclear layer and ganglion cell layer, but there also was a marked increase in the outer-nuclear layer along with general retinal thickness increases in SARS-CoV-2 IN-infected mice (Fig. 2a, c and Supplementary Fig. 5). The increases in the neighbouring retinal layers may be due to extracellular fluid accumulation resulting from the breakdown of the blood-retinal barrier in the deep retinal vascular plexus[49]. We thus assume that both abnormal fluid accumulation within the retina and immune cells infiltration might be involved in the increase in retinal thickness following SARS-CoV-2 infection in this animal model.

Retinal inflammation, diagnosed based on retinitis, can occur following the direct or indirect invasion of pathogens such as cytomegalovirus[50], chikungunya virus[51], and West Nile virus[52]. When retinitis involves the fovea or optic disc, it can exacerbate to reduction or loss of vision[53]. In addition, blurred vision was one of the most common ocular symptoms in patients with COVID-19[2]. Hence, we chose the visual cliff test, which is easy to perform in a BSL-3 facility, to assess the reduction or loss of vision of SARS-CoV-2-infected mice with ocular symptoms (Fig. 3b–d) as the functional consequences of retinal inflammation. Although the number of mice with first foot on the cliff side did not change, the latency to dismount increased in the infected mice with ocular symptoms. We believe that this may be due to the blurring of vision because of ocular discharge. Despite the increased latency, the mice were still able to see the cliff and move toward the safe bench side.

Considering that viral infection, including that caused by SARS-CoV-2, can cause ocular manifestations in humans, several studies have investigated the spread of viruses to ocular tissues. Influenza viruses within the H7 subtype show ocular tropism and use the eyes as an entry route, which was confirmed by administering the virus onto the corneal surface in ferrets[54]. Although Imai et al. introduced a combination of the IN- and ocular routes for SARS-CoV-2 inoculation, they detected an infectious virus in the eyelid tissues of Syrian hamsters[36]. In this study, we examined the dissemination of SARS-CoV-2 to ocular tissues following IN infection and assessed whether the infection route can be reversed by administering the virus to the eyes. Our data suggest unidirectionality of the infectious route, from the lungs to the eyes, in these animal models where ACE2 was expressed in the corneas of the eyes (Supplementary Figs. 9, 10). In particular, the viral burdens of the eye globes were comparatively low and disappeared with time, suggesting the absence of viral proliferation in the eyes (Fig. 4c). Infectious viral particles were not detected in the eyes of Syrian hamsters following ED infection at 6 dpi (Fig. 6d). The prolonged analysis of viral RNA levels following ED infection in the lungs, brain, eye globes, TN, and ON of K18-hACE2 mice and Syrian hamsters at 3, 6, 9 and 12 dpi revealed that the low-level infection of TN and ON at 3 and 6 dpi did not lead to brain infection with time. There was no loss of weight of ED-infected mice until 18 dpi and of ED-infected hamsters until 12 dpi (Supplementary Fig. 11).

This unidirectionality of the infectious route may be due to the formation of the tear film and blinking. The tear film plays an important role in the innate immune system of the eyes for protection against potential pathogens, including lysozyme, lactoferrin, secretory immunoglobulin A, and complement produced in the lacrimal gland[55]. Lysozyme is known to confer anti-HIV activity[56]. Lactoferrin can act as a cationic detergent and disrupt the cell membrane of some bacteria, fungi, and viruses[57]. Secretory immunoglobulin A possibly prevents pathogen adhesion to host cells, blocking further viral infection at mucosal surfaces[58], and is chemotactic for phagocytic neutrophils[59]. Functionally active complement factors in tears are involved in acute inflammatory responses, contributing to innate defence against pathogens[60]. Thus, anti-microbial components in the tear film might provide an immunological and protective environment against viral infections. In addition, the act of blinking provides not only physical protection from outside contaminants, but also helps in the drainage of the tear film from the tear punctum[61].

Clinical studies have reported that the time required for ocular manifestation varied from 15 days to two months after the infection or symptom onset[8,62]. Moreover, Colavita et al. detected viral RNA in the ocular swabs with lower Ct values than those in the nasal swabs of a patient with COVID-19 who suffered from ocular manifestations between 21 and 27 days from the onset of symptoms[63]. Interestingly, they also isolated live replication-competent viruses directly from the ocular fluid collected from the patient. Consistent with the results of these clinical studies, IN- and IT-administered viral copies were detected in the eye globes, which increased in a time-dependent manner (Fig. 4c).

In summary, ocular manifestation and retinal inflammation were promoted by SARS-CoV-2 infection in the mouse model, which increased cytokine production. The virus spreads from the lungs to the brain and eyes through a network consisting of TN and ON. This ocular tropism was also observed in wild-type Syrian hamsters. However, the elicitation of ocular inflammation by a viral infection of the eyes and its clinical relevance remains unknown and warrants further investigation. Along with the respiratory system, eyes and TN should be considered SARS-CoV-2-susceptible organ systems. Our data increases awareness regarding ocular and neuronal infection-mediated disorders beyond respiratory diseases, which will assist in designing treatment strategies for patients with COVID-19.

## Methods

All procedures were performed in a biosafety level 3 (BSL-3) or animal BSL-3 facility for SARS-CoV-2-related experiments, and by personnel equipped with powered air-purifying respirators.

### Animal studies

**Mice.** Eight-week-old male and female B6.Cg-Tg(K18-hACE2)2Prlmn/J mice were purchased from the Jackson Laboratory (Bar Harbor, ME, USA), and 6 to 7-week-old female K18-hACE2 C57BL/6 mice were obtained from the Animal Resource Centre (Perth, Australia). Protocols were approved by the Institutional Animal Care and Use Committee of the Korea Research Institute of Chemical Technology (Protocol ID 8A-M6, IACUC ID 2021-8A-02-01 and 2021-8A-03-03). Immunohistochemistry experiments (Fig. 2c and Supplementary Figs. 2, 5, 10) were approved by the Animal Ethics Committee of Griffith University (MHIQ/08/21) and the procedures conformed to the Australian National Health and Medical Research Council. Mice were maintained under a 12:12 h light/dark cycle at 22–24 °C, with 40–55% humidity and food and water were supplied as desired.

### Syrian hamsters

Eleven-week-old female Golden Syrian hamsters (RjHan:AURA strain) were purchased from Janvier Labs (Saint-Berthevin, France). The hamsters were kept in standard cages of an animal biosafety level 3 (ABL-3) facility and exposed to a 12:12 h light/dark cycle at 22–24 °C, with 40–55% humidity and food and water supplied as desired. Protocols were approved by the Institutional Animal Care and Use Committee of the Korea Research Institute of Bioscience and Biotechnology (KRIBB-ACE-21329, KRIBB-IBC-20220201).

In this study, 25% weight loss was considered the humane euthanasia criterion by $CO_2$ asphyxiation. Organ tissues were collected at the indicated dpi after the animals were anaesthetised using ketamine/xylazine (160 mg/kg ketamine and 16 mg/kg xylazine) or isoflurane in the presence of oxygen in an induction chamber for 5 min, followed by perfusion with 10 ml cold PBS, allowing the blood to flow out. The tissues were weighed and homogenised in preloaded steel bead tubes containing cold PBS using tacoPrep Bead Beater (GeneReach Biotechnology Corp., Taichung City, Taiwan).

### Cells and viruses

The SARS-CoV-2 strain (GISAID Accession ID: EPI_ISL_407193) was propagated in Vero cells (CCL-81, American Type Culture Collection (ATCC), Manassas, VA, USA). The pCC1-4K-SARS-CoV-2-mCherry clone (GenBank Accession No. MT926411) was used for the rescue of infectious virus following the protocol developed by ref. 32. Briefly, 3 μg plasmid DNA was transfected into BHK-21 cells (CCL-10, ATCC) in a six-well plate using Lipofectamine LTX with Plus reagent (15338100, Invitrogen, Waltham, MA, USA) per the manufacturer's instructions. Three days post-transfection, the supernatant was transferred to Vero cells in

a T25 flask. After further incubation for four days, the infectious virus was titrated using a plaque assay.

## Plaque assay

The virus was serially diluted in Eagles minimum essential medium (MEM) supplemented with 2% foetal bovine serum for the plaque assay. The culture medium was removed from the Vero E6 cells in a 24-well plate (~$1 \times 10^5$ per/well) a day before the assay, and the inoculum was transferred onto triplicate cell monolayers. After incubation at 37 °C for 1 h, the inoculum was removed, and the infected cells were overlaid with 1.8% carboxymethyl cellulose in MEM. The samples were incubated for four days, followed by fixation and staining with 0.05% crystal violet containing 1% formaldehyde. The plaques were counted and measured using an ImmunoSpot version 5.0 software and analyser (Cellular Technology Ltd, Shaker Heights, OH, USA).

## Viral inoculations

All viral inoculations (SARS-CoV-2 diluted in PBS, a dose of $10^4$ PFU) were administered via IN, IT, IC, ED and IV routes under anaesthesia using isoflurane or ketamine/xylazine (160 mg/kg ketamine and 16 mg/kg xylazine) in a BSL-3 animal facility, and all efforts were made to minimise animal suffering. IN, IC and IV injections were performed per a previously reported protocol[64]. The mock group was injected with the same volume of PBS in all the experiments. Body weights were measured everyday post-infection.

## IN injection

Mice were infected with twenty microliters inoculum per mouse. The mouse was anaesthetised and the inoculum was administered dropwise into one nostril.

## IT injection

The protocol for IT injection has been described previously[65]. Twenty microliters of the viral suspension was injected per mouse. The anaesthetised mice were placed on the string by their front teeth, with their chest hanging vertically on the platform. The upper chest was illuminated with light of high intensity. The mouth was opened, and the tongue was pulled out with flat forceps to see the white light from the trachea. The catheter was inserted into the trachea, following which, the needle was removed. The inoculum was directly injected into the opening of the catheter. The IT injection was rehearsed using a 2% solution of Evans Blue in normal saline (E2129, Sigma-Aldrich, St. Louis, MO, USA)

## IC injection

Ten microliters of the inoculum was injected per anaesthetised mouse. Before the injection, the inoculum was loaded in a glass microliter syringe (80401, Hamilton, Reno, NV, USA) with a 30 G × 4 mm ultra-fine disposable needle (0J293, Jeongrim Medical, Chungcheongbuk-do, Republic of Korea). The injection site was halfway between the eyes and ears and immediately off the midline. The needle was used to directly and slowly penetrate the cranium. The injection was performed very slowly and was followed by slow removal to prevent efflux. The IC injection was rehearsed using a 2% solution of Evans Blue in normal saline.

## ED

Four microliters of the inoculum was directly administered dropwise onto the corneal surface of both eyes without any scarification, followed by massaging of the eyelids.

## IV injection

Fifty microliters of the inoculum was injected per mouse. The mice were placed in the tail access rodent restrainers, and the inoculum was slowly injected into the tail vein using a 1 ml syringe with a 26 G 1/2 needle (BD, New Jersey, USA).

## Quantitative RT-PCR

Total RNA was extracted from homogenates using Maxwell RSC simplyRNA tissue kit (AS1340, Promega, Madison, WI, USA) following the manufacturer's protocol. Quantitative RT-PCR (QuantStudio 3, Applied Biosystems, Foster City, CA, USA) was performed using a one-step Prime script III RT-qPCR mix (RR600A, Takara, Kyoto, Japan). The viral RNA of NP was detected using a 2019-nCoV-N1 probe (10006770, Integrated DNA Technologies, Coralville, IA, USA).

## Multiplex analysis

The eyes of SARS-CoV-2-infected mice were dissected at 0, 3, and 6 dpi, and then homogenised in bead tubes (a-psbt, GeneReach Biotechnology, Taichung, Taiwan). Aliquots were analysed using the MILLIPLEX human cytokine/chemokine magnetic bead panel (HCYTOMAG-60K, Merck Millipore, Burlington, MA, USA) using the Luminex 200 multiplexing instruments (40-012, Merck Millipore) to assess cytokine/chemokine expression.

## H&E staining

The protocol for mouse eye sections has been described previously[66]. The eyes, including the appendages, were collected and fixed overnight in Davidson's fixative (BBC Biochemical, Mount Vernon, WA, USA). The fixed eyes were excised to balance the pressure and preserve the morphology. They were then processed routinely and embedded in paraffin wax (ASP300S, Leica, Wetzlar, Germany). Subsequently, the eyes were cut into 7-µm retinal cross sections and stained with H&E (BBC Biochemical) using an auto-stainer (ST5010, Leica). Images were obtained using an Olympus BX51 microscope (Olympus, Tokyo, Japan). The retinal thickness was measured using the Nuance 3.02 software (PerkinElmer, Waltham, MA, USA).

## Immunofluorescence staining

Cryosectioning and immunofluorescence staining[67] were performed. K18-hACE2 mice were sacrificed via $CO_2$ asphyxiation or intraperitoneal injection of ketamine/xylazine on day 6 post-infection. The eyes, including appendages, were collected and fixed using 4% paraformaldehyde in PBS for 2 h or overnight. After overnight incubation in 20% sucrose or in 30% sucrose for 6 h at 4 °C, they were transferred to a cryomold for embedding in optimal-cutting-temperature (OCT) compound (AGR1180, Agar Scientific, Essex, UK) and frozen at −80 °C overnight. Ten micrometre retinal cross sections obtained from the OCT frozen tissue block were washed thrice with wash buffer (0.5% Tween 20 in PBS) for 5 min to remove OCT. Then, the sections were incubated with blocking buffer (2% bovine serum albumin (BSA), 10% normal goat serum in PBS) for 1 h, followed by further incubation overnight at 4 °C with antibodies against spike protein (40150-T62-COV-2, Sino Biological, Beijing, China; 1:100), γ-synuclein (sc-65979 AF488, Santa Cruz Biotechnology, Dallas, TX, USA; 1:50), Gr-1 (550291, BD Biosciences, Franklin Lakes, NJ, USA; 1:300), CD3 (100366, BioLegend, San Diego, CA, USA; 1:350), CD4 (550280, BD Biosciences; 1:400), CD8 (550281, BD Biosciences; 1:400), human ACE2 (ab15348, Abcam, Cambridge, UK; 1:350), and ACE2 (MAB933, R&D Systems, Minneapolis, MN, USA; 1:100) in wash buffer containing 1% BSA. After washing, the primary antibodies were detected using anti-rat or anti-rabbit Alexa-Fluor 488, AlexaFluor 568, AlexaFluor 647 (Thermo Fisher Scientific, Waltham, MA, USA; 1:1000), AlexaFluor 594 (Life Technologies, Carlsbad, CA, USA; 1:1000) or streptavidin DyLight650 (Thermo Fisher Scientific). Nuclei were stained with DAPI (62247, Thermo Fisher Scientific, Waltham, MA, USA). The slides were mounted with ProLong Gold antifade agent (Thermo Fisher). Immunofluorescence was observed using confocal microscopy (LSM700, Carl Zeiss, Oberkochen,

Germany). Z-stacks (13 slices) were imaged and merged to form a maximum-intensity projection. Some images were acquired using a confocal microscope (Olympus FV3000, Olympus, Tokyo, Japan) at $1024 \times 1024$ resolution using 20× and 40× (with 2x zoom) objectives and processed using FV31S-DT (Olympus built-in software).

## Visual cliff test

A visual cliff test was performed based on the protocol developed by ref. 68 with some modifications. The mice were tested in an open-topped acrylic glass box ($40 \times 30 \times 50$ cm). The light source in the experimental room was dimmed (about 20 lx). The barriers were opaque to prevent reflection. A paper with a large black/white chequered pattern ($2 \times 2$ cm$^2$) was placed under one half of the plate ('bench side'), while the bottom of the other half ('cliff side') was covered with a small chequered pattern sheet ($1 \times 1$ cm$^2$) to emphasise the cliff drop-off. The mice were placed on the black central platform ($0.4 \times 30 \times 0.1$ cm) between the bench side and the cliff side, and their activities were recorded for 2 min to measure the latency to dismount and the direction of the first foot on the bench or cliff sides.

## In vivo fluorescence spectrum unmixing

SARS-CoV-2-mCherry-infected mice were sacrificed at 6 dpi and perfused with 4% paraformaldehyde in PBS. The lungs, brain, and other organs were immediately collected to detect fluorescent signals. Using the IVIS Lumina S5 system (PerkinElmer, Wrentham, MA, USA), the tissues were sequentially imaged with a fixed emission filter at 520 nm to determine the optimal excitation from 420 to 480 nm (1 s, F-stop = 1, medium binning). The images were analysed using the Living Image 4.7 software (PerkinElmer, Wrentham, MA, USA). The epi-fluorescence was expressed in units of radiant efficiency [p/sec/cm$^2$/sr]/[μW/cm$^2$] after subtracting the background signal.

## Statistical analysis

All experiments were performed at least two times. All data were analysed using the GraphPad Prism 8.0 software (GraphPad Software, San Diego, CA, USA). Data were shown as means with the standard error of mean (SEM). For experiments with only two groups, an unpaired two-tailed $t$-test was used. One-way analysis of variance (ANOVA) was used for experiments with three and more groups. $P < 0.05$ was considered statistically significant. Specific analytical methods are described in the figure legends.

## Reporting summary

Further information on research design is available in the Nature Portfolio Reporting Summary linked to this article.

# Data availability

All source data generated in this study are provided with this article. Source Data are provided with this paper as a Source Data file. Source data are provided with this paper.

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

## Acknowledgements

This work was supported by grants from the National Research Council of Science & Technology (NST) funded by the Korean government (MSIP; CRC-16-01-KRICT to Y.-C.K.), (MSIT; CRC21021, KRIBB KGM5242221 to H.-J.K.), National Research Foundation of Korea (NRF) funded by the Ministry of Education, Science, Technology (MSIT) of the Korean government (2020R1C1C1003379 to Y.-C.K.) and the Australian National Health and Medical Research Council (NHMRC) Grant (APP1184879 to S.M). S.M. is the recipient of the NHMRC Senior Research Fellowship (APP1154347).

## Author contributions

G.U.J. and Y.-C.K. conceived of and S.M., N.J.C.K. and Y.-C.K. supervised this study. G.U.J., H.-J.K., S.M. and Y.-C.K. designed experiments. G.U.J., H.-J.K., W.H.N., X.L., Z.L.L., A.G.S., N.J.C.K., A.T., H.W.M., G.Y.Y., H.-J.S., I.-C.L. and D.-G.A. performed experiments. G.U.J., J.S.C., S.M. and Y.-C.K. analysed data. H.-J.K., S.-J.K., S.M. and Y.-C.K. provided essential reagents. G.U.J., H-J.K., S.M. and Y.-C.K. wrote the manuscript. S.M. and Y.-C.K. are joint senior authors. All authors reviewed and edited the manuscript.

## Competing interests

The authors declare no competing interests.
