## [Peer Review File · Nature Communications]

Reviewer comments, first round review -

Reviewer #1 (Remarks to the Author):

The article "Ocular tropism of SARS-CoV-2 with retinal inflammation through neuronal invasion in animal models" aims to understand how SARS-CoV-2 affects the orbit and whether this could be a route of infection of the virus. The authors use well established models to study this phenomenon with interesting approaches to the question, but the data presented is unclear in its ability to support the claims presented. In order to be considered for publication in Nature Communications, we believe this manuscript will require a significant amount of work before it can be ready.

Major points

- Although spike protein is localized at the level of the GCL, the retinal thickness increase is occurring at locations beyond the GCL. In this case, what is the driving force of this? Cells that are already there? CD45+ immune cell infiltration? Fluid build up? Please show experimental data of what is occurring. Also curious that the histology in 2A shows clear increases in thickness of the retina, but IF in figure 1 F do not seem to demonstrate much change in increases of the thickness of the retina. Is this true when you look at IF images across many mice?
- In evaluating the functional consequences of the retinal inflammation; K-18 mice infected with SARS-CoV-2 in our experience (and others) have significant lethargy, lack of grooming behaviors and other signs of severe morbidity before they succumb to death as seen by the authors around day ~8-10. It is surprising that they are able to perform these tests as well as non infected mice. Would a better test to check the activity of their retinas be to use an ERG instead?
- In the diverse injection routes; we understand the argument for saying that the Trigeminal and Optic nerve routes are the likely routes affecting both the brain and eyes through the experiments; but why is it the case that eye drops will still have infection present in the optic nerve and trigeminal nerve but will not see any infection eventually in the brain? Why would the infectious route have a one way directionality in this case? It is lower than other organs, but once it gets there, does it not have the ability to proliferate to high levels like it does in the brain? The authors comment on this, but would be helpful to know experimentally/or have more of a detailed hypothesis on why this is the case.
- It is unusual to see such a robust infection in the brain in hamsters (not really noticed in other groups studies). Are these infectious virus (plaque)? What do you think accounts for the differences in these experimental conditions versus previous published literature?
- In general, more information regarding why this unidirectionality exists in both mice and hamsters would add a lot to the current manuscript. In addition, what cell types are being infected? Is the transmission through neurons, other cell types in these tissues? Detailed immunofluorescence of cell types infected in the nerves will help clarify this.

Minor points

- First paragraph in introduction does not add onto manuscript/current story, can be omitted.
- will be helpful to know whether nerves harbor infectious virus
- interesting that the concentration of virus is higher proximal to the eyes in figure 4D and not closer to the brain. Could the authors comment on why this might be if the directionality is coming from the brain to the eyes?

Reviewer #2 (Remarks to the Author):

In this study, the authors show that intranasal inoculation of hACE2 mice with SARS-CoV-2 leads

to detectable infectious virus in the eye, causing histopathological changes to the retina and induction of proinflammatory cytokine and chemokine responses post-inoculation in ocular tissue. Authors employ fluorescently labeled virus to show that viral spread occurs from the respiratory tract to the eyes via the brain, identifying the optic nerve and trigeminal nerve as key factors supporting this spread. In contrast, inoculation of hACE2 mice (and Golden Syrian hamsters) by dropwise virus administration to the eye surface did not result in robust infection, supporting that these models are better suited to demonstrate extrapulmonary spread from SARS-CoV-2 from the respiratory tract to eye and are not suitable models to study ocular exposure to the virus. Studies investigating multiple exposure routes to SARS-CoV-2 virus are lacking in the literature, and inclusion of two mammalian models in this study is a benefit. The study appears generally complete and well-controlled, though there are places throughout the manuscript where text needs to be modified to best reflect the conclusions supported by the data presented. Overall, additional contextualization of results and specificity of language is needed for this study to be suitable for publication.

Major comments:

-Lines 119-121, authors employ intratracheal inoculation to demonstrate that infectious virus spreads from the lungs to optic nerves while bypassing the lacrimal duct, but do not provide sufficient data to justify this claim. Authors do not show nasal tissue virus replication in this study; it is possible that intratracheal inoculation could nonetheless result in nasal titers post-inoculation (following replication in lungs), thus spreading to the eye via the lacrimal duct. Authors should modify language to better specify IT inoculation results support their conclusions but (in the absence of confirming a lack of upper respiratory tract replication in these animals) cannot rule out other avenues of virus spread.

-Lines 129-137, authors do not discuss the suitability of hACE2 mice (or Syrian hamsters employed later in the study) for ocular exposure to SARS-CoV-2. Are permissive receptors for the virus present on the ocular surface of these transgenic mice or hamsters? This should be discussed and contextualized throughout the study; discussing antiviral factors in tear fluid in the discussion is helpful but (without experimental data to support their role) incomplete to understand what is occurring. As such, the statement on line 194-5 ("...we certainly excluded the possibility of ocular surface as a portal of entry for the virus") is too generalized to be substantiated; authors should reword to specify their findings are specific to the models employed in this study and cannot be immediately applied to other mammals.

-Authors should better contextualize their findings in the discussion with other studies in the literature which have examined spread of SARS-CoV-2 to ocular tissues (both mammalian models and in humans). In example, Imai et al show detectable virus in eyelid tissue of some Golden Syrian Hamsters following joint nasal/ocular inoculation (reference 38 in this submission), as have some studies employing intranasally inoculated ferrets. The discussion would benefit in general from enhanced contextualization in general; the results from the visual cliff test are not discussed, and interpretation of retinal inflammation are not well-linked to other studies in mammalian models or humans.

-It is unclear why authors report converting TCID50 titration values to PFU, especially as they are citing a study from 1972 to provide a conversion formula (lines 257-9). Viruses can differ broadly between titration substrates; if the authors intend to convert virus titers from one unit to another, this conversion must be determined for the strain employed in the study, not by citing a study that did not employ a SARS-CoV-2 virus.

-Data in Figure 4D nicely shows detection of fluorescently labeled virus in the optic nerve, but not the globe of the eye itself. However, it is unclear from the text if collection of ocular tissue for viral titer and cytokine analysis was limited to the globe of the eye or included residual optic nerve material – this must be clearly stated to help the reader understand if data presented in figures 1 and 2 reflects material from the globe itself or is inclusive of globe and optic nerve both.

-Authors demonstrate a capacity for SARS-CoV-2 to spread to the eye following intranasal inoculation, supporting a tropism of SARS-CoV-2 for ocular tissue, but this tropism is not

maintained following ocular inoculation. Suggest modifying the title to include specifying intranasal inoculation to better reflect the findings of this study (e.g. "...through neuronal invasion in animal models following intranasal inoculation").

Minor comments:

-The methods section is lacking relevant detail. Importantly, the authors must specify the humane endpoint criteria employed in this study, especially as Figure 4A shows weight loss that appears to exceed 30% of baseline. The method of euthanasia of mice for necropsy is not specified. Additionally, please specify the gauge needle used for IV injection (lines 290-1). The method of tissue homogenization prior to titration should be specified. And clarity is needed for why tissues from mice are reported as PFU/g (Fig 1D) whereas tissues from hamsters are reported as PFU/ml (Fig 5C).

-What was the limit of detection for titration assays? This should be included in the methods and the graphs should be revised accordingly to reflect this LOD (e.g. figure 1D, negative tissues should be set at the LOD, not 0, as it not likely the actual LOD is 0 log PFU/g).

-Line 150, please employ more technical language than "...and then pulled the brain out."

-Lines 193-4, the study cited reports scarifying the mouse eye prior to dropping virus on the surface (whereas this work omitted a scarification step); suggest authors revise language to more clearly indicate that the experimental methods employed between their work and this published study are not equivalent.

-Specify day p.i. tissues were harvested in Figure 1D-E.

-References on lines 257 and 259 appear written in text but are not included in the references section.

REVIEWER COMMENTS

Reviewer #1 (Remarks to the Author):

The article “Ocular tropism of SARS-CoV-2 with retinal inflammation through neuronal invasion in animal models” aims to understand how SARS-CoV-2 affects the orbit and whether this could be a route of infection of the virus. The authors use well established models to study this phenomenon with interesting approaches to the question, but the data presented is unclear in its ability to support the claims presented. In order to be considered for publication in Nature Communications, we believe this manuscript will require a significant amount of work before it can be ready.

Response: We thank the reviewer for his/her consideration and helpful comments in improving our manuscript. We have responded to each of the critiques and incorporated changes in the appropriate sections of the revised manuscript. A point-by-point discussion of the comments from the reviewer is provided below.

Major points

- Although spike protein is localized at the level of the GCL, the retinal thickness increase is occurring at locations beyond the GCL. In this case, what is the driving force of this? Cells that are already there? CD45+ immune cell infiltration? Fluid build up? Please show experimental data of what is occurring. Also curious that the histology in 2A shows clear increases in thickness of the retina, but IF in figure 1 F do not seem to demonstrate much change in increases of the thickness of the retina. Is this true when you look at IF images across many mice?

Response: We thank the reviewer for this suggestion. Retinal and choroidal thickness have been evaluated as potential inflammatory markers for inflammatory diseases (reference 22). In patients with ocular Behcet’s disease, one of the inflammatory diseases mentioned before, the eyes showed moderate inflammation and marked inflammatory cell infiltration in the retinae, implicating vasculitis and haemorrhage (DOI: 10.1016/S0039-6257(97)00026-X, DOI: 10.3109/08916939209148460). Notably, a recent clinical study revealed that patients with COVID-19 showed lesions at the ganglion cell layer (GCL) and inner plexiform layer (IPL) in both eyes (DOI: 10.1016/S0140-6736(20)31014-X). In addition, another clinical study reported that the patients presented flame-shaped haemorrhages along the retinal vascular arcades and peripheral retinal haemorrhages (DOI: 10.1136/bjophthalmol-2020-317576). The breakdown of the blood-retinal barrier can cause abnormal fluid build-up within the retina following increase in retinal thickness (DOI: 10.1001/archophth.124.2.193). This breakdown might induce infiltration of more immune cells, promoting inflammatory responses. To identify the infiltrating immune cells, we performed immunofluorescence assay by staining the eye sections with anti-CD3, anti-CD4, anti-CD8 antibodies for T cells, and anti-GR-1 antibody for neutrophils. Higher T cell and neutrophil levels were observed in the eyes of SARS-CoV-2-infected mice at 6 dpi than in mock-infected mice (Fig. 2c, Supplementary Fig. 5). We thus believe that both abnormal fluid accumulation within the retina and immune cell infiltration are involved in the increase in retinal thickness following SARS-CoV-2 infection in this animal model.

Fig. 2c

Supplementary Fig. 5

As observed with H&E staining, distinct increase in retinal thickness was observed in mice in IHC and immunofluorescence. Figure 1f shows that more cells stained with DAPI in ONL, INL, and GCL in the eyes of SARS-CoV-2-infected mice than those of mock-infected mice, which was evident even with naked eyes. We have also shown other representative images below to easily appreciate the differences in thickness between mock- and SARS-CoV-2-infected mice.

- In evaluating the functional consequences of the retinal inflammation; K-18 mice infected with SARS-CoV-2 in our experience (and others) have significant lethargy, lack of grooming behaviors and other signs of severe morbidity before they succumb to death as seen by the authors around day ~8-10. It is surprising that they are able to perform these tests as well as non infected mice. Would a better test to check the activity of their retinas be to use an ERG instead?

Response: We thank the reviewer for this suggestion. As the reviewer mentioned above, SARS-CoV-2-infected K18 mice show lethargy at 8–10 days post infection (dpi). Therefore, we performed the visual cliff test early at 5 dpi. Although the mice showed reduced mobility at 5 dpi, they still felt aversion to depth and were able to move to resolve it. They also felt uncomfortable, which motivated them to move to the narrow width and low height of the black central platform ($0.4 \times 30 \times 0.1$ cm) where they were first placed. We have enclosed representative videos of this test. The infected mice with ocular discharge might have blurred vision as reported in patients with COVID-19 (reference 2). Increased latency to dismount may be due to the blurred vision. Although it took longer, they were able to see the cliff and move toward the safe bench side.

The electroretinogram (ERG) is a sensitive and noninvasive method for testing retinal function by measuring the electrical response of retinal cells to a light stimulus (DOI: 10.21769/BioProtoc.2218). This test can be utilized to assess retinal degeneration (DOI: 10.1167/iops.02-0438). As this test requires expertise and equipment in a BSL-3 facility, such as the Ganzfeld Colour Dome, the Espion visual electrophysiology system, and two Bayer-Mittag contact lens electrodes, we chose a simple test “the visual cliff test” instead of ERG to assess retinal degeneration or visual loss, which has been verified by several studies. (DOI: 10.1016/j.scr.2015.08.007; DOI: 10.1371/journal.pone.0200417; DOI: 10.1093/hmg/ddg249; DOI: 10.1016/j.scr.2015.08.007). Retinal inflammation involving the fovea or optic disc exacerbates reduction or loss in vision (reference 49). Hence, we believe that the visual cliff test can be easily performed in a BSL-3 facility and is appropriate to evaluate the functional consequences of retinal inflammation. We have now specifically mentioned these points in the Discussion (lines 206-215).

- In the diverse injection routes; we understand the argument for saying that the Trigeminal and Optic nerve routes are the likely routes affecting both the brain and eyes through the experiments; but why is it the case that eye drops will still have infection present in the optic nerve and trigeminal nerve but will not see any infection eventually in the brain? Why would the infectious route have a one way directionality in this case? It is lower than other organs, but once it gets there, does it not have the ability to proliferate to high levels like it does in the brain? The authors comment on this, but would be helpful to know experimentally/or have more of a detailed hypothesis on why this is the case.

Response: We appreciate the suggestion offered here. To investigate whether low infection of the optic nerve and trigeminal nerve eventually leads to the brain infection, we have performed an additional experiment, where we performed prolonged analysis of viral RNA levels following eye drop infection in the lungs, brain, eye globes, trigeminal nerve, and optic nerve of mice and hamsters at 3, 6, 9, and 12 dpi (Supplementary Fig. 11). The body

weight of mice did not change following eye drop infection until 18 dpi. The low infection in nerves did not lead to brain infection in these animal models over time. The comparatively low and disappearing viral loads in the eye globes suggested that viral proliferation in the eyes was not high (Fig. 4c). There were no infectious viral particles in the eyes of Syrian hamsters following ED infection at 6 dpi (Fig. 6d). We believe that these results firmly support the unidirectionality of the infectious route in these animal models. In addition to these experiments, we suggested a detailed hypothesis to explain the unidirectionality of infection. The anti-microbial components in the tear film, including lysozyme, lactoferrin, secretory immunoglobulin A, and complement produced in the lacrimal gland, might provide an immunological and protective environment against viral infection. We have now comprehensively discussed those points and added references in the fourth paragraph of the Discussion (lines 234-243).

- It is unusual to see such a robust infection in the brain in hamsters (not really noticed in other groups studies). Are these infectious virus (plaque)? What do you think accounts for the differences in these experimental conditions versus previous published literature?

Response: We thank the reviewer for this comment. As the reviewer mentioned, a previous paper reported that virus titres in the brains in Syrian hamsters were lower than those in the lungs following inoculation with $10^{5.6}$ PFU in 110 μ l via a combination of the intranasal (100 μ l) and ocular (10 μ l) routes (reference 36). To address this comment, we have repeated the experiments with more Syrian hamsters ($n = 7$). As before, we inoculated Syrian hamsters with 10^4 PFU in 100 μ l via the intranasal route and then assessed viral RNA levels at 6 dpi in the lungs, brain, eye globes, optic nerves, trigeminal nerves, and spleen using RT-qPCR, and obtained similar results (the accumulated results are shown below; $n = 11$), indicating that the viral loads in the brain of IN-infected Syrian hamsters were higher than those in the lungs. A newly performed plaque assay at 6 dpi demonstrated that these viruses in the brains were infectious, although the difference in infectious viral load between the lungs and the brains was not as much as the differences in virus RNA levels. These results are likely similar to those of other groups that showed that infectious virus was not detected at 7 dpi despite the continued detection of high copies of viral RNA in the lungs of SARS-CoV-2-infected Syrian hamsters (DOI: 10.1038/s41586-020-2342-5). We believe that the detection of higher copies of virus RNA in the brains than those in the lungs may account for the ACE2 expression pattern in Syrian hamsters. *Ace2* mRNA expression was higher in the brains than in the lungs, supporting viral replication and transmission (DOI: 10.3389/fphar.2020.579330). However, we were not able to understand why this result differed from those of other studies except that the strain and source of Syrian hamsters and the way of inoculation varied.

- In general, more information regarding why this unidirectionality exists in both mice and hamsters would add a lot to the current manuscript. In addition, what cell types are being infected? Is the transmission through neurons, other cell types in these tissues? Detailed immunofluorescence of cell types infected in the nerves will help clarify this.

Response: As stated above in the response to the reviewer’s third comment, we have now comprehensively discussed a detailed hypothesis on the unidirectionality in the fourth paragraph of the Discussion (lines 234-243).

To investigate the cell types in the eyes that were infected, we performed IHC and immunofluorescence as shown below (Supplementary Fig. 3). GCL, where spike proteins were detected in Fig. 1f, is mainly composed of the nucleus of the retinal ganglion cells (RGCs; DOI: 10.1007/s00415-019-09654-w). In addition, a recent study of human retinal organoids revealed that SARS-CoV-2 infects RGCs (DOI: 10.1016/j.stemcr.2022.02.015). γ -Synuclein and RBPMS are two well-characterized RGC markers (γ -synuclein: reference 21; RBPMS: DOI: 10.1002/cnc.23521). We tested several antibodies for detecting these proteins using IHC-immunofluorescence; however, only anti- γ -synuclein antibody (sc-65979 AF488, Santa Cruz) worked. As expected, co-stained cells in GCL with anti-spike and anti- γ -synuclein antibodies were observed at 6 dpi, indicating that the infected cells were RGCs. We have mentioned this in the revised manuscript (lines 67-69).

Supplementary Fig. 3

Minor points

-First paragraph in introduction does not add onto manuscript/current story, can be omitted.

Response: We thank you for this suggestion. As suggested, we have removed it in the revised manuscript.

-will be helpful to know whether nerves harbor infectious virus

Response: We appreciate this suggestion. We should have assessed whether trigeminal and optic nerves harbour infectious viral particles, and not simply viral RNA levels, to support our conclusion. The infectious viral particles of the trigeminal and optic nerves were detected using plaque assay, and their pattern was similar to those of the virus RNA levels (lines 120-122; Supplementary Fig. 8).

Supplementary Fig. 8

-interesting that the concentration of virus is higher proximal to the eyes in figure 4D and not closer to the brain. Could the authors comment on why this might be if the directionality is coming from the brain to the eyes?

Response: We apologise for the confusion because of using a font size that was too small, and this issue has been addressed in the revised manuscript. The fluorescence in the brain ($\times 10^9$ units) was higher than that in the eyes ($\times 10^6$ units), supporting the directionality from the brain to the eyes. This is now clearly shown in Fig. 5d.

Reviewer #2 (Remarks to the Author):

In this study, the authors show that intranasal inoculation of hACE2 mice with SARS-CoV-2 leads to detectable infectious virus in the eye, causing histopathological changes to the retina and induction of proinflammatory cytokine and chemokine responses post-inoculation in ocular tissue. Authors employ fluorescently labeled virus to show that viral spread occurs from the respiratory tract to the eyes via the brain, identifying the optic nerve and trigeminal nerve as key factors supporting this spread. In contrast, inoculation of hACE2 mice (and Golden Syrian hamsters) by dropwise virus administration to the eye surface did not result in robust infection, supporting that these models are better suited to demonstrate extrapulmonary spread from SARS-CoV-2 from the respiratory tract

to eye and are not suitable models to study ocular exposure to the virus. Studies investigating multiple exposure routes to SARS-CoV-2 virus are lacking in the literature, and inclusion of two mammalian models in this study is a benefit. The study appears generally complete and well-controlled, though there are places throughout the manuscript where text needs to be modified to best reflect the conclusions supported by the data presented. Overall, additional contextualization of results and specificity of language is needed for this study to be suitable for publication.

Response: We thank the reviewer for his/her consideration and helpful comments in improving our manuscript. We have tried to improve the contextualization of the revised manuscript by toning down the strong conclusions related to tissue tropism and infection routes as suggested by the reviewer. A point-by-point discussion of the reviewer's comments is provided below.

Major comments:

-Lines 119-121, authors employ intratracheal inoculation to demonstrate that infectious virus spreads from the lungs to optic nerves while bypassing the lacrimal duct, but do not provide sufficient data to justify this claim. Authors do not show nasal tissue virus replication in this study; it is possible that intratracheal inoculation could nonetheless result in nasal titers post-inoculation (following replication in lungs), thus spreading to the eye via the lacrimal duct. Authors should modify language to better specify IT inoculation results support their conclusions but (in the absence of confirming a lack of upper respiratory tract replication in these animals) cannot rule out other avenues of virus spread.

Response: We agree with this reviewer's concern that progeny virus post-IT inoculation may move to the eyes through the nasolacrimal duct following viral replication in the lungs. We realized that IT inoculation cannot rule out the spread of the virus to the eyes via the nasolacrimal duct. Hence, we have revised sentences to specify that IT infection only is responsible for the dissemination of the virus from the lungs to the eyes and brain (lines 122-123). We have also added this reviewer's concern as a limitation of this study in the Discussion (lines 199-205).

-Lines 129-137, authors do not discuss the suitability of hACE2 mice (or Syrian hamsters employed later in the study) for ocular exposure to SARS-CoV-2. Are permissive receptors for the virus present on the ocular surface of these transgenic mice or hamsters? This should be discussed and contextualized throughout the study; discussing antiviral factors in tear fluid in the discussion is helpful but (without experimental data to support their role) incomplete to understand what is occurring. As such, the statement on line 194-5 ("...we certainly excluded the possibility of ocular surface as a portal of entry for the virus") is too generalized to be substantiated; authors should reword to specify their findings are specific to the models employed in this study and cannot be immediately applied to other mammals.

Response: We thank the reviewer for pointing these out. Previous studies have reported that ACE2 and TMPRSS2 are expressed on human (reference 12, 14) and murine (reference 30) ocular surfaces. To investigate whether

ACE2 is expressed on the ocular surface of both K18-hACE2 mice and Syrian hamsters, we performed immunofluorescence staining for ACE2 in the eyes of these animal models as shown below. We have now added these references and new data in the Results and Discussion (lines 134, 224-225; Supplementary Fig. 9 and 10). In addition, we have reworded the text to indicate that our findings are specific to these animal models, and have toned down our conclusion in the fourth paragraph of the Discussion.

Supplementary Fig. 9

Supplementary Fig. 10

-Authors should better contextualize their findings in the discussion with other studies in the literature which have examined spread of SARS-CoV-2 to ocular tissues (both mammalian models and in humans). In example, Imai et al show detectable virus in eyelid tissue of some Golden Syrian Hamsters following joint nasal/ocular inoculation (reference 38 in this submission), as have some studies employing intranasally inoculated ferrets. The discussion would benefit in general from enhanced contextualization in general; the results from the visual cliff test are not discussed, and interpretation of retinal inflammation are not well-linked to other studies in mammalian models or humans.

Response: We thank the reviewer for mentioning these critical points that we had missed. We have now revised the text to comprehensively discuss these points in the Discussion. We have referred to other studies that examined the viral spread following ocular inoculation to rationalise the ED route (lines 216-221). Furthermore, we have now specifically discussed the results from the visual cliff experiment and interpreted the retinal inflammation with additional citations (lines 206-215). The increase in the latency to dismount may be due to the blurring of vision, which is one of the most common ocular symptoms of patients with COVID-19 (reference 2). Although the time to dismount was delayed, the infected mice with ocular symptoms were still able to see the visual cliff and move toward the safe bench side. Retinal inflammation, diagnosed based on retinitis, can occur due to infection by cytomegalovirus (reference 46), chikungunya virus (reference 47), and West Nile virus (reference 48). The retinal inflammation in response to viral infection can be exacerbated to reduction or loss in vision if the retinitis involves the fovea or optic disc (reference 49). This is why we chose the visual cliff test to assess vision as a functional consequence of retinal inflammation.

-It is unclear why authors report converting TCID50 titration values to PFU, especially as they are citing a study from 1972 to provide a conversion formula (lines 257-9). Viruses can differ broadly between titration substrates; if the authors intend to convert virus titers from one unit to another, this conversion must be determined for the strain employed in the study, not by citing a study that did not employ a SARS-CoV-2 virus.

Response: We appreciate and agree with the reviewer's suggestion. To address this comment and eliminate the confusion, we again performed the plaque assay to analyse virus titres in the lungs and eyes, including appendages, of mock- or SARS-CoV-2-infected Syrian hamsters. The results are shown in Fig. 6c and 6d, and the manuscript is now appropriately revised (lines 173-176).

-Data in Figure 4D nicely shows detection of fluorescently labeled virus in the optic nerve, but not the globe of the eye itself. However, it is unclear from the text if collection of ocular tissue for viral titer and cytokine analysis was limited to the globe of the eye or included residual optic nerve material? This must be clearly stated to help the reader understand if data presented in figures 1 and 2 reflects material from the globe itself or is inclusive of globe and optic nerve both.

Response: We thank the reviewer for pointing this out. The collection of ocular tissues for viral titre and cytokine analysis in Fig. 1d, 1e, and 3a included residual optic nerve material, which are ocular appendages. We have now

clearly stated this point throughout the revised manuscript, distinguishing it from the eyes, including appendages, and the eye globes (lines 56, 85, 114, 118, 120, 124, 127, 138, 142, 154, 174, 175, 179, 351, and 362).

-Authors demonstrate a capacity for SARS-CoV-2 to spread to the eye following intranasal inoculation, supporting a tropism of SARS-CoV-2 for ocular tissue, but this tropism is not maintained following ocular inoculation. Suggest modifying the title to include specifying intranasal inoculation to better reflect the findings of this study (e.g. "...through neuronal invasion in animal models following intranasal inoculation").

Response: Thank you for this suggestion. We have modified the title in the revised manuscript.

Minor comments:

-The methods section is lacking relevant detail. Importantly, the authors must specify the humane endpoint criteria employed in this study, especially as Figure 4A shows weight loss that appears to exceed 30% of baseline. The method of euthanasia of mice for necropsy is not specified. Additionally, please specify the gauge needle used for IV injection (lines 290-1). The method of tissue homogenization prior to titration should be specified. And clarity is needed for why tissues from mice are reported as PFU/g (Fig 1D) whereas tissues from hamsters are reported as PFU/ml (Fig 5C).

Response: We thank the reviewer for this comment. We have now revised Fig. 5a (lines 152-153) and have added more details in the Materials and Methods section in terms of the anaesthesia method used for necropsy, humane endpoint criterion, the needle gauge used for IV injection, and tissue homogenisation in lines 280-285 and 336-337.

We have used 25% weight loss as the humane euthanasia criterion by CO₂ asphyxiation in our protocol. Organ tissues were collected at the indicated dpi after the animals were anesthetized using ketamine/xylazine (160 mg/kg ketamine, 16 mg/kg xylazine) or isoflurane in the presence of oxygen in an induction chamber for 5 min, followed by perfusion with 10 ml cold PBS, allowing the blood to flow out. As this was the first time we investigated pathogenesis following infection using SARS-CoV-2-mCherry viruses, the weight changes of the infected mice were monitored to the end. But, we have now re-adjusted the weight loss criterion to 25% in the revised manuscript.

The reason why the units for the viral titration differed between mice and hamsters was because of differences in the methods used for mice (plaque assay) and hamsters (TCID₅₀). To eliminate the confusion, we have re-performed the viral titration using plaque assay for hamsters. The manuscript and figures have been appropriately revised (lines 173-176; Fig. 6c, 6d).

-What was the limit of detection for titration assays? This should be included in the methods and the graphs should be revised accordingly to reflect this LOD (e.g. figure 1D, negative tissues should be set at the LOD, not 0, as it not likely the actual LOD is 0 log PFU/g).

Response: We tried to detect plaques from spleen of both SARS-CoV-2-infected mice and Syrian hamsters as negative control to set the LOD similar to that used for the assessment of viral RNAs. We did not detect any plaque from the spleen even at the 10^{-1} dilution or no dilution. Thus, we believe that the LOD for plaque assay is 0 log PFU/g.

-Line 150, please employ more technical language than "...and then pulled the brain out."

Response: We apologise for using non-academic language, and have now modified sentences using more technical language (lines 155-157). "For detecting the fluorescence in the TN and ON of euthanized mice, we cut the parietal bone along the left and right sides and the sagittal suture, followed by removal of the brain gently to expose the base of the cranial cavity (Fig. 5c)."

-Lines 193-4, the study cited reports scarifying the mouse eye prior to dropping virus on the surface (whereas this work omitted a scarification step); suggest authors revise language to more clearly indicate that the experimental methods employed between their work and this published study are not equivalent.

Response: We thank the reviewer for pointing this out. We have withdrawn the citation and have substituted it with a research paper showing that the virus inoculum was administered dropwise to the surface of the right eye of each ferret and massaged using the eyelid without scarifying (reference 50; lines 218-219). In this study, we inoculated the virus in the same way by dropping it onto the corneal surface of both eyes without scarifying, followed by massaging with the eyelid (lines 332-333).

-Specify day p.i. tissues were harvested in Figure 1D-E.

Response: We appreciate for bringing this to our notice. We have now stated 'days post infection (dpi)' throughout the revised manuscript.

-References on lines 257 and 259 appear written in text but are not included in the references section.

Response: We thank the reviewer for pointing this out. We have corrected this in the revised manuscript.

Reviewer comments, second round review -

Reviewer #1 (Remarks to the Author):

We thank the authors for trying to answer all the comments with more detail and additional citations.

Two additional thoughts include:

- 1) The reason for the swelling is even more unclear with the data presented. The immune infiltrates are not in the regions where there is expansion of the layers- not providing clear evidence of the pathophysiology.
- 2) The hypothesis behind the unilateralism is nice, but to be frank, makes this reviewer more confused about what the phenotypes in the retina represent. There seems to be great inconsistency between how the eye drops affect the retina and other nerves and the phenotypes that are being described here.

Reviewer #2 (Remarks to the Author):

In this revised manuscript, authors have performed extensive revisions to the original submission, providing several needed control experiments and additional text-based modifications to address study limitations, overall tempering conclusions drawn. Specifically, authors have performed additional immunohistochemistry work and virologic assessments to assess the presence of infectious virus, and provided several additional sections to the discussion to more thoroughly contextualize findings with the literature and specify potential limitations with the study design. This has strongly improved the overall quality of the submission and suitability of the manuscript for publication. Authors addressed comments raised during the first round of review to my satisfaction; minor additional comments remain.

Minor comments:

Lines 225-6, why is the function of the ocular surface as a portal of entry for virus “unlikely”?

Lines 234-243, how much is known regarding the composition of tear film in the mouse and hamster models employed in this study relative to humans? Do these species have comparable fluidics exchange between ocular and respiratory tissues as humans?

Lines 269-273, please specify the institution for which IACUC approval was conferred, and clarify what authors mean by “some animal experiments” which were further approved by Griffith University.

REVIEWER COMMENTS

Reviewer #1 (Remarks to the Author):

We thank the authors for trying to answer all the comments with more detail and additional citations.

Two additional thoughts include:

1) The reason for the swelling is even more unclear with the data presented. The immune infiltrates are not in the regions where there is expansion of the layers- not providing clear evidence of the pathophysiology.

Response: We thank the reviewer for this comment. As we described in the previous response for the retinal swelling, the breakdown of the blood-retinal barrier (BRB) can cause immune cells infiltration and abnormal fluid accumulation within the retina following increase in retinal thickness. A previous study of increased retinal thickness suggested that there could be no direct correlation between the regions of increased retinal thickness and the sites of the BRB breakdown (DOI: 10.1001/archophth.118.10.1364). This discrepancy could be explained by the preferential pathways for fluid movement in the retina, by the possibility of varying rates of retinal fluid reabsorption, and by the previous BRB breakdown sites that had stopped at the moment of the examination. We thus believe that BRB breakdown and retinal thickening may occur simultaneously, either in association with each other or independently. Another study reported the highest increases in retinal thickness in INL accompanied by the increases in the neighboring layers ranged from GCL to ONL (DOI: 10.1159/000438792). The authors in that study suggested that the increases of the neighboring retinal layers' thickness may be due to extracellular fluid accumulation resulting from alteration of the BRB in the deep retinal vascular plexus. We have now included an additional section for the retinal swelling with additional citations in the Discussion (lines 206-217).

Since neutrophils and T cells infiltration was detected in INL and GCL as shown in Fig. 2c and Supplementary Fig. 5, we next performed the single-cell imaging mass cytometry using Hyperion Imaging System to specify detailed cell types localized in thickened retinal regions in SARS-CoV-2-infected mice. While the lung and other tissues could be labeled with about thirty cell-type specific antibodies, unfortunately, we were not able to label cell types in the eyes with technical problems, possibly owing to differences in cellular density and diversity.

2) The hypothesis behind the unilateralism is nice, but to be frank, makes this reviewer more confused about what the phenotypes in the retina represent. There seems to be great inconsistency between how the eye drops affect the retina and other nerves and the phenotypes that are being described here.

Response: We thank the reviewer for this comment. We have suggested that the unidirectionality of virus spread is from the lungs to the eyes, but not eyes to the brain or the lungs. As described in the manuscript and previous responses to reviewers, when we infected mice and hamsters with SARS-CoV-2 via eye drops, disappearing viral loads and no infectious viral particles were detected in the lungs, brain, eye globes, trigeminal nerve, and optic nerve until 12 days post infection (Fig. 4c, 6d, 6f, and Supplementary Fig. 11). Consequently, there was no ocular symptom or weight loss in these animal models following eye drop infection. Our data have indicated that the intranasal (IN) infection caused the phenotypes in the retina and ocular symptoms through viral spread via trigeminal nerve and optic nerve. We believe that these results strongly support the unidirectionality of the infectious route from the lungs to eyes, but not the opposite.

Reviewer #2 (Remarks to the Author):

In this revised manuscript, authors have performed extensive revisions to the original submission, providing several needed control experiments and additional text-based modifications to address study limitations, overall tempering conclusions drawn. Specifically, authors have performed additional immunohistochemistry work and virologic assessments to assess the presence of infectious virus, and provided several additional sections to the discussion to more thoroughly contextualize findings with the literature and specify potential limitations with the study design. This has strongly improved the overall quality of the submission and suitability of the manuscript for publication. Authors addressed comments raised during the first round of review to my satisfaction; minor additional comments remain.

Minor comments:

Lines 225-6, why is the function of the ocular surface as a portal of entry for virus “unlikely”?

Response: We apologise for the confusion by using an unnecessarily long sentence. To address this comment and eliminate the confusion, the manuscript is now appropriately revised (lines 237).

Lines 234-243, how much is known regarding the composition of tear film in the mouse and hamster models employed in this study relative to humans? Do these species have comparable fluidics exchange between ocular and respiratory tissues as humans?

Response: We thank the reviewer for this comment. Although the composition of tear film and fluidics exchanges between ocular and respiratory tissues in humans are well-known, those in mice and hamsters still need uncovered. A recent comparative analysis of tear composition in humans, domestic mammals, reptiles, and birds revealed that there is relatively high similarity in the biochemical components between humans, dogs, and horses (DOI: 10.3389/fvets.2020.00283). However, to the best of our knowledge, there is no comparative analysis regarding the composition of tear film and fluidics exchanges between humans, mice, and hamsters.

Lines 269-273, please specify the institution for which IACUC approval was conferred, and clarify what authors mean by “some animal experiments” which were further approved by Griffith University.

Response: Thank you for this suggestion. Because this is the double-blind review, we did not specify the institutions for which IACUC approvals. We will correctly specify them in the manuscript before the publication. In addition, we have now stated what animal experiments are approved by Griffith University (lines 282-283).

Reviewer comments, third round review -

Reviewer #1 (Remarks to the Author):

Thank you for answering these questions. Although the responses do not let us get to the bottom of why some of these phenomenon occur, I hope that we will get answers in future experimental studies!